# The Appeal and Reality of Recycling LoRAs with Adaptive Merging

Haokun Liu [* 1 2]  Gyung Hyun Je [* 1 2]  Marco Ciccone [2]  Zhenlin Xu [3]  Prasanth YSS [4]  Colin Raffel [1 2 5]

## Abstract

The widespread availability of fine-tuned LoRA modules for open pre-trained models has led to an interest in methods that can adaptively merge LoRAs to improve performance. These methods typically include some way of selecting LoRAs from a pool and tune merging coefficients based on a task-specific dataset. While adaptive merging methods have demonstrated improvements in some settings, no past work has attempted to recycle LoRAs found "in the wild" on model repositories like the Hugging Face Hub. To address this gap, we consider recycling from a pool of nearly 1,000 user-contributed LoRAs trained from the Llama 3.1 8B-Instruct language model. Our empirical study includes a range of adaptive and non-adaptive merging methods in addition to a new method designed via a wide search over the methodological design space. We demonstrate that adaptive merging methods can improve performance over the base model but provide limited benefit over training a new LoRA on the same data used to set merging coefficients. We additionally find not only that the specific choice of LoRAs to merge has little importance, but that using LoRAs with randomly initialized parameter values yields similar performance. To better understand why past work has proven successful, we confirm that positive transfer is indeed possible when there are highly relevant LoRAs in the pool. We release the model checkpoints and code online at https://github.com/r-three/realistic-adaptive-merging.

## 1 Introduction

Open pre-trained language models (Grattafiori et al., 2024; Team et al., 2025; Yang et al., 2025) combined with the

---

[1]University of Toronto, Canada [2]Vector Institute, Toronto, Canada [3]Mistral AI (work done before joining) [4]Layer 6 AI [5]Hugging Face. Correspondence to: Haokun Liu <haokunliu412@gmail.com>, Gyung Hyun Je <jayje@cs.toronto.edu>.

*Proceedings of the 43rd International Conference on Machine Learning*, Seoul, South Korea. PMLR 306, 2026. Copyright 2026 by the author(s).

efficiency and effectiveness of training parameter-efficient adapter modules like LoRA (Hu et al., 2022) have made it remarkably cheap and easy to fine-tune language models. Such fine-tuning is often data-efficient (Liu et al., 2022; Je & Raffel, 2025; Aghajanyan et al., 2021), producing a performant model on a target task even if only a few dozen examples are available. These adapters are frequently shared on model repositories like the Hugging Face model hub, where, for example, over 1,000 fine-tuned variants of the popular Llama 3.1 8B-Instruct model (Grattafiori et al., 2024) are available. The widespread public availability of LoRAs has motivated research into *recycling* LoRAs to improve performance on new, unseen tasks (Huang et al., 2024; Yang et al., 2024; Wu et al., 2023). A common building block in these recycling methods is model merging (Wortsman et al., 2022; Ilharco et al., 2023b; Yadav et al., 2023; Matena & Raffel, 2022), where the parameters of constituent fine-tuned models are combined to produce a single model that (hopefully) retains the capabilities of the constituent models. One common way of recycling LoRAs to improve target-task performance is via "adaptive merging", where the per-model coefficients used during merging are tuned on a supervised task-specific dataset (Yang et al., 2024; Huang et al., 2024; Chronopoulou et al., 2023).

Despite the rapidly growing interest in methods for recycling LoRAs (Yadav et al., 2025b), to the best of our knowledge, no past work has considered collections of user-contributed LoRAs found "in the wild". Instead, past work generally evaluates methods on a set of bespoke LoRAs that are trained through controlled fine-tuning on a curated set of target tasks. **Our primary aim in this study is to address this gap by using a pool of LoRAs directly recycled from user contributions to the Hugging Face Hub.** This pool not only represents a much greater diversity in terms of datasets, training methods, and hyperparameters than has been considered in past work, but is also dramatically larger. Therefore, we seek answers for the following questions:

1. How effective is merging recycled LoRAs in realistic, data-constrained settings?

2. When limited target-task data is available, to what extent can data-dependent adaptive merging of recycled LoRAs meaningfully outperform simple fine-tuning?

We confirm that many of these recycled LoRAs, both used

in isolation and merged with a selected pool of LoRAs, can indeed improve the base model's performance on a diverse range of downstream tasks. To ensure the reliability of results, we formalize a design space of adaptive merging methods and perform a thorough ablation study to uncover new methodological combinations. However, we highlight that adaptive merging methods typically tune per-LoRA merging coefficients using a small labeled target-task dataset, which can also be used to train a task-specific LoRA (as done in past works (Wu et al., 2023)). We find that adaptive merging methods fail to produce meaningful and consistent improvements when compared to this target-task LoRA.

Through further analysis, we uncover that when the target-task LoRA is included in the merging pool, choosing a *random* set of recycled LoRAs for merging works comparably to more sophisticated selection methods that choose LoRAs based on parameter-space similarity or each LoRA's performance on the target task. To better understand this behavior, we additionally perform adaptive merging with a pool of LoRAs with *randomly sampled* parameter values, ultimately finding that the performance remains similar. This suggests that the benefits of adaptive merging over the target-task LoRA may primarily stem from a regularization effect, instead of "positive transfer" (Ruder, 2019; Pruksachatkun et al., 2020; Vu et al., 2020) from the recycled LoRAs. Since past research on adaptive merging has produced promising results, we complete our study by considering a pool of bespoke, consistently trained LoRAs. We intentionally construct a pool of LoRAs that are highly relevant to the studied downstream tasks and confirm that positive transfer occurs in this unrealistic (but nevertheless common) setting. Taken together, our results elucidate the benefits and shortcomings of adaptive merging methods and provide a rigorous foundation for future work on recycling LoRAs.

## 2  Background

Although large language models can competently perform a growing range of tasks, they often fall short on specialized tasks or domains (Kandpal et al., 2022; Zhu et al., 2025; Je & Raffel, 2025), e.g., when private knowledge or skills are involved (Challapally et al., 2025). Often, performance on target tasks can be improved via fine-tuning, which consequently remains a widespread practice. Here, we provide background on adaptation methods, model merging, and a high-level overview of the merging methods studied in our work.

### 2.1  Parameter-efficient fine-tuning (PEFT)

PEFT methods (Han et al., 2024) have emerged as a valuable tool for adapting large pre-trained models to downstream tasks. These methods keep the majority of the base model parameters frozen during training and only update or add a small number of parameters, reducing the memory cost from gradient and optimizer states and facilitating aggressive quantization of the base model (Dettmers et al., 2023). By limiting the number of trainable parameters, PEFT has also been shown to mitigate catastrophic forgetting (Biderman et al., 2024), improving transfer learning and performance in few-shot settings (Liu et al., 2022). Collectively, these properties have been instrumental in democratizing fine-tuning of frontier models, making it accessible for users with limited computational infrastructure and data.

**Low-Rank Adaptation (LoRA)** (Hu et al., 2022) is the most widely adopted PEFT method. LoRA selects a set of target weight matrices from all the linear projection layers within the Transformer architecture and reparameterizes the update of each weight $W \in \mathbb{R}^{n \times m}$ as a low-rank product:

$$W_{\text{finetuned}} = W_{\text{pretrained}} + sBA, \quad (1)$$

where $A \in \mathbb{R}^{r \times m}$, $B \in \mathbb{R}^{n \times r}$ are trainable parameters, $r \ll \min(m, n)$ is the rank hyperparameter, and $s$ is a scaling factor which is generally dependent on $r$, to make the optimal learning rate more stable. A key advantage of this formulation is that the $sBA$ update can be folded directly to the original weights for inference, thereby simplifying computation and allowing direct integration with advanced serving tools and optimized kernels (Kwon et al., 2023; Ollama Team, 2023; Hsu et al., 2025) designed for the standard Transformer architecture. By choosing which linear projections to update and changing the rank, LoRA can be tuned to maximize performance.

### 2.2  Merging methods

Merging is the process of combining multiple constituent models, usually sharing the same architecture, into a new model. Merging task-specific models aims to produce a multitask model that is competent on all of the constituent models' tasks. It sometimes yields improved generalization compared to multitask learning (Tam et al., 2024a). Merging also enables decentralized training pipelines with multiple branches of model development in parallel (Cohere et al., 2025). Merging's reuse of specialized models is synergistic with PEFT modules built on top of open models, which has led to the active exploration of merging algorithms.

**Simple averaging** (McMahan et al., 2017; Wortsman et al., 2022) is a simple but effective merging method:

$$W_{\text{merged}} = \sum_{i=1}^{k} \alpha_i W_i \quad (2)$$

where $k$ is the number of models to merge, $W_i$ represents the parameters of the $i$-th model, and $\alpha_i$ are the merging coefficients, usually set to $1/k$.

**TIES Merging** (Yadav et al., 2023) operates on task vectors (Ilharco et al., 2023a), i.e., the difference between the

fine-tuned and pre-trained parameter values ($sBA$ in the case of LoRA). TIES first trims redundant parameters by retaining only those with the largest magnitude changes, then resolves sign disagreements by choosing the dominant sign across models for each parameter, and finally averages only the parameters that agree with the elected sign. This approach mitigates destructive interference that occurs when task vectors point in opposing directions.

**Task Singular Vectors (TSV) Merging** (Gargiulo et al., 2024) leverages the observation that task-specific knowledge in fine-tuned models can be localized to a low-dimensional subspace. By performing singular value decomposition on task vectors, TSV identifies the principal directions that capture task-relevant adaptations. Merging is then performed in this reduced space, which helps preserve task-specific information while reducing interference.

**Arrow** (Ostapenko et al., 2024) routes per-token hidden state to the most relevant LoRA in the pool at every layer zero-shot, relevance determined by the hidden state alignment with each LoRA's representation.

### 2.3 Adaptive merging methods

Adaptive merging methods tune the weight assigned to each model (e.g., $\alpha_i$ in simple averaging) to maximize performance on a target task. In this work, we focus on methods that tune static merging coefficients (i.e., coefficients that do not vary based on the input token) on a labeled dataset. For a broader discussion of "MoErging" methods that route inputs to different adapter modules, see Yadav et al. (2025a).

**LoraHub** (Huang et al., 2024) is an adaptive merging framework that composes multiple LoRAs for new tasks. Given a collection of LoRAs trained on diverse tasks, LoraHub randomly selects a set of LoRAs, assigns a combination coefficient to each LoRA, and merges the LoRAs accordingly. The coefficients are optimized on few-shot examples from the target task using a gradient-free blackbox optimization algorithm, enabling rapid adaptation to novel tasks by reusing existing LoRAs without backpropagation through the base model.

$\pi$-**Tuning** (Wu et al., 2023) is a targeted merging method that leverages task similarity to enhance transfer learning. It first computes task embeddings using the Fisher Information Matrix (FIM), which captures the sensitivity of the model to parameter perturbations for each task. Given a target task, $\pi$-Tuning identifies the most similar tasks with the embedding, interpolates similar LoRA together with a new randomly initialized LoRA, and jointly optimizes the LoRAs and interpolation coefficients during training.

**AdaMerging** (Yang et al., 2024) assigns a single learnable coefficient $\lambda_k$ to each task vector, while Layer-wise AdaMerging learns separate coefficients for each task vector

of each weight matrix, enabling more fine-grained control. Its variant AdaMerging++ incorporates TIES merging.

## 3 Our Framework

*Table 1.* Adaptive merging configurations in our framework.

| | **Design choice** | | | |
|---|---|---|---|---|
| **Method** | **Selection** | **Tuning** | **Granularity** | **Activation** |
| AdaMerging | Random | Grad-based | Module | Linear |
| $\pi$-tuning | Quasi-FIM | Joint | Module | Softmax |
| LoraHub | Random | Grad-free | Model | Linear |
| **Ours** | Evaluation | Grad-based | Module | Leaky ReLU |

We propose a unified framework that captures the common design elements across adaptive merging methods, with the exception of methods that dynamically vary coefficients per-token or per-example through learned routing mechanisms, as they require additional inference-time computation. This framework helps us systematically explore various merging approaches and tailor existing methods to our problem setting. We explore the following design decisions in adaptive merging methods:

**1) Selection**: it determines how to choose a subset of $k$ LoRAs from a pool of $N$ available LoRAs, where typically $k \ll N$. This step is necessary for adaptive merging methods that load all selected LoRAs into memory at once, which is infeasible for large $k$. ***Random***: Uniformly samples $k$ LoRAs without replacement, ignoring task relevance. ***Evaluation***: Ranks LoRAs by their accuracy on the target task training set and selects the top $k$. ***Cosine***: Ranks LoRAs based on their parameter-space cosine similarity to a target-task LoRA. ***Clamp***: Similar to *Cosine*, but clamps negative elements to zero when computing the inner product, accounting for merging methods that discard conflicting parameters. ***Quasi-FIM***: Past work has used the FIM as a notion of task similarity (Achille et al., 2019; Vu et al., 2020). Rigorous diagonal Fisher Information Matrix estimation requires examples from each LoRA's training dataset. When recycling LoRAs, we only have access to the LoRAs themselves and not the datasets used to train them. We therefore approximate FIM by treating each LoRA as a single (pseudo-) gradient step, and therefore computing cosine similarity on their elementwise squares (Li et al., 2025).

**2) Granularity**: it determines how and where merging coefficients are learned, trading off expressiveness against optimization complexity. ***Model***: A single scalar $\alpha_i$ applied uniformly across all parameters ($k$ coefficients) in each LoRA. ***Layer***: Separate coefficients $\alpha_i^{(l)}$ for each layer ($k \times L$ coefficients). ***Sublayer***: Separate coefficients $\alpha_i^{(l)}$ for each sublayer (attention and feedforward separately) ($k \times L \times 2$ coefficients). ***Module***: Separate coefficients $\alpha_i^{(l,m)}$ for each module type (e.g., query, key, value in self-attention matrices) within each layer.

**3) Coefficient Activation**: it determines how the raw merging coefficients are transformed before applying them to LoRA weights. ***Softmax***: Normalizes coefficients across LoRAs to form a probability distribution, ensuring $\sum_i \alpha_i = 1$ with $\alpha_i \geq 0$. ***Leaky ReLU***: Allows primarily positive coefficients while permitting small negative values, as prior work shows performance degrades when transitioning from interpolation to extrapolation (Gueta et al., 2023). ***Linear***: Applies no transformation, leaving coefficients unconstrained.

**4) Tuning:** it determines how the merging coefficients and, optionally, LoRA parameters are optimized. ***Gradient-Based***: Optimizes merging coefficients via gradient descent on a target-task objective. ***Gradient-Free***: Optimizes coefficients using black-box optimization algorithms that do not require backpropagation through the model. ***Joint***: Simultaneously optimizes both the merging coefficients and the underlying LoRA parameters on the target task.

Our preliminary experiments reveal that tuning *module-level* merging coefficients with *evaluation-based* LoRA selection works best. Gradient-based coefficient tuning outperforms the gradient-free alternative. Leaky ReLU coefficient activation with zero initialization provides a modest improvement. Results for other configurations are detailed in Section E.

Concretely, we select the top-$k$ LoRAs based on their target-task evaluation scores. For the target module $m$ at layer $l$ with pretrained weight $W^{(l,m)}$, let $A_i^{(l,m)}$ and $B_i^{(l,m)}$ denote the corresponding low-rank matrices of the $i$-th selected LoRA, $s \in \mathbb{R}^k$ the LoRA scaling coefficients, and $\alpha^{(l,m)} \in \mathbb{R}^k$ the learnable merging coefficients, where $\alpha_i^{(l,m)}$ is the coefficient for the $i$-th LoRA. The merged update and output for the input $x$ at this target module are then given by:

$$\tilde{\alpha}^{(l,m)} = \text{LeakyReLU}(\alpha^{(l,m)})$$

$$\hat{W}^{(l,m)} = \sum_{i=1}^{k} s_i \, \tilde{\alpha}_i^{(l,m)} \, A_i^{(l,m)} B_i^{(l,m)} \qquad (3)$$

$$y = W^{(l,m)} x + \hat{W}^{(l,m)} x$$

In the experiments that follow, unless stated otherwise, we use this recycling method and denote it as "Ours" in tables and figures. We also implement the adaptive merging methods from §2 within this framework (Table 1).

## 4 Recycling LoRAs in the Wild

Past research into recycling LoRAs has frequently been motivated by the observation that there is an ever-growing collection of user-contributed LoRAs models available on public hubs (Horwitz et al., 2025). Reusing this wealth of available public LoRAs is an appealing prospect, as it would allow recycling already expended compute while ideally transferring knowledge and capabilities from the

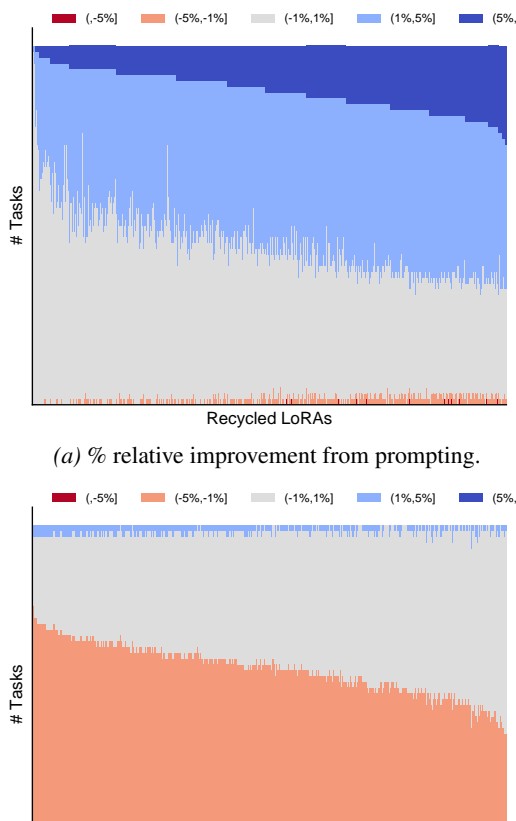

*(a) % relative improvement from prompting.*

*(b) % relative improvement from target-task LoRA.*

*Figure 1.* Rescaled LoRA performance across 62 downstream tasks, categorized into five buckets based on their relative % improvement over prompting or LoRA baselines. LoRAs are sorted along the x-axis by their performance.

diverse data sources used by individual contributors. However, to the best of our knowledge, past studies on recycling LoRAs have not used publicly available user-contributed ones, but have instead trained a bespoke collection on curated datasets. A primary aim of our work is to consider using adaptive merging to recycle from a "realistic" pool of user-contributed modules. This motivates the question we explore in this section: *Do publicly available, user-contributed LoRAs actually contain useful knowledge for downstream target tasks?*

### 4.1 Recycled LoRA Pool Overview

At the time of writing, the Hugging Face Hub hosts hundreds of thousands of LoRAs[1], each trained on top of a particular base model. We focus on LoRAs based on Llama 3.1 8B-Instruct (Grattafiori et al., 2024) (or quantized vari-

---

[1] https://huggingface.co/models?library=peft

ants thereof), which had the most LoRAs publicly available at the time of experimentation. We forgo all the LoRAs that do not have license information or use safetensor[2]. We include LoRA (Hu et al., 2022) and rsLoRA (Kalajdzievski, 2023) modules trained on causal language modeling tasks, and leave out other PEFT methods or task types like embedding. We also skipped models that use advanced LoRA features like rank pattern, alpha pattern, and training particular indices in the vocabulary. There are occasional practical issues from the remaining models, e.g., parameter shape inconsistency with the stated base model, abnormally large or NaN values in weights; we remove these corner cases. The final filtered pool has **958 LoRAs**. Unlike past work, we highlight that this collection is 1) highly heterogeneous, with modules varying in their ranks and the specific transformer blocks they target (Section A), 2) completely user-contributed with no models trained by us, and 3) an order of magnitude larger than any pool considered in past work.

## 4.2 Recycled LoRA Quality Assessment

To investigate whether recycling LoRAs from our pool improves performance on downstream target tasks, we first assess whether they individually encode useful knowledge.

**Evaluation setup** We build on top of a collection of 62 target tasks from Je & Raffel (2025) and Tam et al. (2024b) as our testbed. These tasks have desirable characteristics for our purposes: they span multiple domains from science to natural language understanding and include task types such as multiple-choice question-answering and open-ended generation; they have varying difficulty levels, with Llama 3.1 8B-Instruct's performance lagging significantly behind human-level on certain tasks; and they benefit from fine-tuning to various degrees, with some tasks reaching human-level performance from fine-tuning on relatively few target-task examples and others requiring thousands before performance plateaus. For more details, see the discussion in Je & Raffel (2025) and Section B. We assume access to 100 samples from the target task, 80 for training and 20 for validation, fixed across all the experiments. Unless specified otherwise, we report the performance on the test set.

**Experiment Design** As a first test of whether capabilities can transfer from our recycled pool, we tune the scaling coefficient on each recycled LoRA individually for each downstream task. We tune the coefficients at module-level granularity using leaky ReLU, which is the best configuration we found in our exploration in §3. As a baseline, we also train a target-task LoRA for each task using the 100 available samples, repeating for 5 random seeds (see Section C.2 for details). We then report the performance on the test set, comparing the tuned recycled LoRA against the base model and the target-task LoRA performance.

---

[2] https://huggingface.co/docs/safetensors/

**Results** Figure 1a shows that several recycled LoRAs do encode useful knowledge for downstream tasks. In particular, the top LoRAs achieve more than 5% relative improvement (dark-blue bars) compared to the base model on several tasks, while most achieve 1-5% relative improvement across many tasks (light-blue bars). However, few recycled LoRAs reach meaningfully higher performance than the target-task LoRA (Figure 1b), highlighting the strength of simple fine-tuning with a few downstream data points. Availability of relevant LoRAs also varies considerably across tasks: certain tasks have a subset of highly relevant recycled LoRAs with notable positive transfer, whereas other tasks achieve little to no improvement from any selected LoRA from the pool (Section B). Overall, these results confirm that positive transfer is possible, but suggest that reliable selection of useful LoRAs is necessary and that some target tasks may lack relevant public LoRAs entirely.

## 5 Realistic Evaluation of Adaptive Merging

We now turn to evaluating merging methods using our pool of recycled LoRAs. We take the non-adaptive methods (Simple Averaging, TIES Merging, and TSV Merging) and the adaptive methods (LoraHub, $\pi$-Tuning, AdaMerging, and Arrow) introduced in §2, and evaluate their ability to learn each target task with the help of 958 hub LoRAs. In addition to existing methods, we report the performance of our adaptive merging method from §3 ("Ours"), using the best-performing configuration found through our extensive search over the merging, training, and selection design space.

**Methods** Although non-adaptive methods do not require LoRA selection, they are rarely applied at this many LoRAs. Therefore, we limit the pool to 30 randomly selected LoRAs (performance averaged over three seeds). In addition, we include a simple averaging baseline over all LoRAs. We implement the adaptive methods listed in Table 1, following their original implementations when possible. In cases where a method requires resources unavailable in our problem setting—e.g., $\pi$-Tuning uses training data of the recycled LoRAs to calculate the FIM—we find close alternatives within our framework. All adaptive methods select 30 LoRAs using their corresponding selection method except for $\pi$-Tuning, which selects 20 instead, as it has more trainable parameters and therefore demands more GPU memory. We assume limited access to 100 target-task samples for any data-dependent optimization (e.g., tune and select merging coefficients, select LoRAs based on target-task performance). More implementation details can be found in Section C.1 and Section D.

**Target-Task LoRA** Adaptive merging methods generally leverage target-task data to tune merging coefficients, implying that a target-task training set is available. Consequently, there is no actual barrier to training a LoRA directly on this

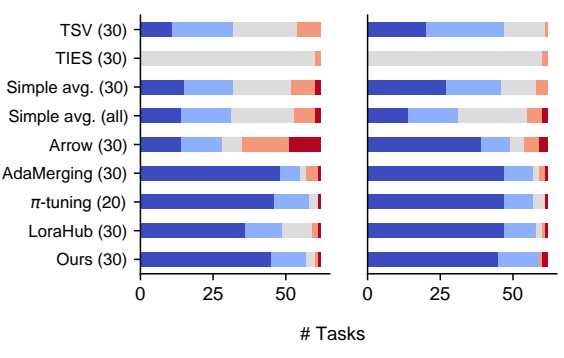

*(a) % acc. improvement compared to* **prompting**, *without (left) and with (right) target-task LoRA.*

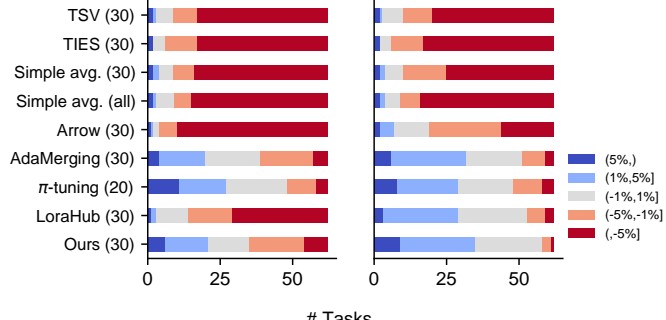

*(b) % acc. improvement compared to* **LoRA baseline**, *without (left) and with (right) target-task LoRA.*

*Figure 2.* Avg. % acc. improvement from baselines across all 62 tasks, without and with the target-task LoRA in the merging pool. TSV, TIES, Simple Averaging, and Arrow are non-adaptive methods, while AdaMerging, $\pi$-Tuning, LoraHub, and Ours are adaptive methods. The number in parenthesis (e.g., TSV (30)) indicate the number of merged LoRAs.

dataset instead. However, this possibility has often been overlooked in past works (Huang et al., 2024; Pari et al., 2024). To fill this gap, for each task, we train a target-task LoRA using the same 100 training and validation examples used to determine the merging coefficients, and explicitly study the utility of including it in the pool of candidate LoRAs.

**Ablations on model family and sample size** In §5, we report the results using Llama 3.1 8B-Instruct as the base model and 100 target-task samples. To further study the impact of base model family and available sample size on merging performance, we replicate all experiments under two new settings: i) using Qwen/Qwen3-4B-Instruct-2507 (Team, 2025) with 1956 LoRAs post-filtering, and ii) using a lower target-task budget (10 samples) for tuning merging coefficient and target-task LoRA. We discuss the results in detail in Section F.

**Q1. Does recycled LoRA merging improve the base model?**

To analyze whether merging recycled LoRAs improves over zero-shot prompting of the base model, we run merging on recycled LoRAs both with and without the target-task LoRA in the pool, reporting the percentage improvement relative to zero-shot prompting. The raw average downstream task performance by method is found in Table 2.

As Figure 2a shows, compared to the base model, adaptive merging using recycled LoRAs can achieve notable improvement on many downstream tasks. **Non-adaptive merging methods** that use fixed uniform coefficients underperform adaptive methods that tune merging coefficients. Substituting the randomly selected LoRAs with evaluation-based LoRAs fails to close this gap (Section F.3). Simple averaging over all LoRAs performs notably worse than selecting 30 LoRAs at random, indicating that even though there is no computational obstacle to including all LoRAs,

doing so is counterproductive. As another outlier, TIES shows minimal improvement over the prompting baseline. Through extensive hyperparameter search on TIES (pruning percentage and per-LoRA weight), we find that performance using recycled LoRAs is highly sensitive to hyperparameter choices and LoRA composition, and its best hyperparameter setting still underperforms other methods. We suspect TIES, originally designed for full fine-tuning, is difficult to apply to LoRA-trained models. A large number of LoRAs from unknown sources further exacerbates the challenge. See Section D for further discussion. Among **adaptive merging methods**, LoraHub is slightly less performant than others; its gradient-free tuning or coarser granularity may explain this gap. Given that our exploration also highlights the effect of granularity in this scenario, we hypothesize the latter is primarily responsible.

Including the target-task LoRA improves performance for most methods. It bridges the gap between LoraHub and other adaptive merging methods, but doesn't affect $\pi$-Tuning or TIES merging meaningfully.

| Method | Average accuracy | |
|---|---|---|
| | w/o LoRA | w/ LoRA |
| TSV (30) | 0.488 | 0.522 |
| TIES (30) | 0.467 | 0.466 |
| Simple avg. (30) | 0.496 | 0.536 |
| Simple avg. (all) | 0.495 | 0.496 |
| Arrow (30) | 0.471 | 0.613 |
| Adamerging (30) | 0.652 | 0.669 |
| $\pi$-Tuning (20) | 0.668 | 0.668 |
| LoraHub (30) | 0.563 | 0.663 |
| Ours (30) | 0.652 | 0.675 |
| Prompting (zero-shot) | 0.467 | |
| LoRA | 0.657 | |

*Table 2.* Average performance across 62 downstream tasks. "LoRA" here refers to a LoRA trained on target-task data.

## Q2. Does recycled LoRA merging outperform the LoRA baseline?

The target-task LoRA significantly boosts performance when included in the pool of LoRAs; however, this observation also demands a more stringent evaluation criterion. Adaptive merging is essentially a PEFT method that can accept recycled LoRAs as supplementary resources — it is only meaningful insofar as it at least outperforms simply training a LoRA on the target task.

We show in Figure 2b and Table 2 that merging is substantially less effective when compared to the target-task LoRA baseline. Methods that do not optimize merging coefficients mostly remain below the target-task LoRA's performance across almost all downstream tasks, even when the target-task LoRA is included in the merging pool. In contrast, adaptive merging methods show clear benefits over the target-task LoRA when it is included, achieving mostly comparable performance otherwise. As before, LoraHub is particularly dependent on access to the target-task LoRA; without it, the method produces almost no positive transfer. $\pi$-Tuning exhibits little performance difference with or without the target-task LoRA. Since $\pi$-Tuning includes an untrained LoRA (randomly initialized A matrix with B set to zero) in the pool that is jointly optimized with the specialized LoRAs, we hypothesize that training this LoRA during merging coefficients tuning has a similar effect to including a pre-trained target-task LoRA. AdaMerging and our method perform similarly well.

Our results demonstrate that for adaptive merging methods, including the target-task LoRA is almost always necessary. Moreover, once the target-task LoRA is included, performance differences across design choices largely diminish.

## Q3. Does the selection strategy meaningfully impact the merging process?

Once the target-task LoRA is included in the pool, different adaptive merging methods exhibit similar performance trends despite employing vastly different selection mechanisms. To investigate, we conduct a more thorough experiment varying the number of selected LoRAs, focusing on two selection strategies: evaluation-based and random. Intriguingly, Figure 3 shows that although evaluation-based selection clearly outperforms random when the target-task LoRA is not used, indicating carefully selected LoRAs form a better space to update the model, the performance gap becomes negligible when target-task LoRA is included.

If selecting random LoRAs from the pool works as well as a more informed selection strategy, what knowledge is actually being transferred? And how random can the LoRAs be while still retaining these benefits? To answer this question, we consider the extreme case where the "recycled" LoRAs are set to random values. We repeat the experiment using

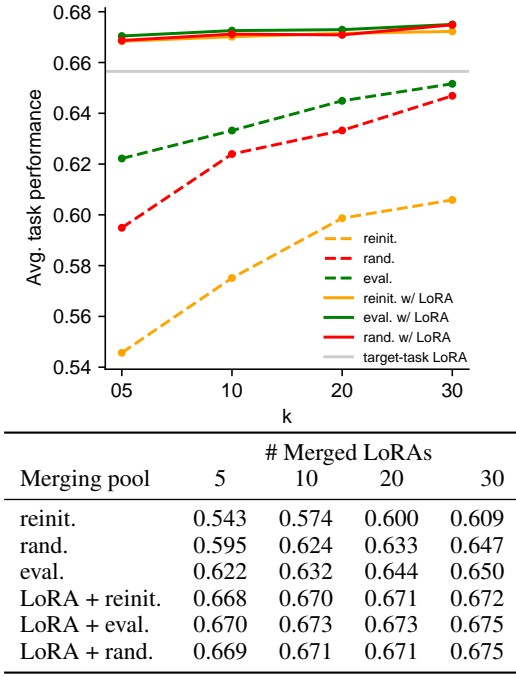

| | # Merged LoRAs | | | |
| Merging pool | 5 | 10 | 20 | 30 |
| --- | --- | --- | --- | --- |
| reinit. | 0.543 | 0.574 | 0.600 | 0.609 |
| rand. | 0.595 | 0.624 | 0.633 | 0.647 |
| eval. | 0.622 | 0.632 | 0.644 | 0.650 |
| LoRA + reinit. | 0.668 | 0.670 | 0.671 | 0.672 |
| LoRA + eval. | 0.670 | 0.673 | 0.673 | 0.675 |
| LoRA + rand. | 0.669 | 0.671 | 0.671 | 0.675 |

*Figure 3.* Adding the target-task LoRA significantly reduces the impact of LoRA selection method. The average LoRA baseline performance across tasks is shown as a gray line.

our adaptive merging method, selecting $k \in \{5, 10, 20, 30\}$ LoRAs but reinitializing them from a normal distribution, with the standard deviation matched to that of the original LoRA A and B matrices respectively. The target-task LoRA is kept intact. Figure 3 confirms that these randomly initialized LoRAs perform comparably to the actual recycled LoRAs across all values of $k$, as long as the target-task LoRA is in the pool. This suggests that, with the target-task LoRA present, the gains from adaptive merging may stem primarily from a regularization effect rather than task-relevant knowledge transfer from recycled LoRAs.

To test our regularization effect hypothesis, we compare the gains in performance between target-task LoRA with regularization technique applied during training and adaptive merging with randomly initialized LoRAs. We test different dropout rates {0.2, 0.3, 0.5}, weight decay values {0.01, 0.1}, and batch sizes {2, 4} and report the improvement over the LoRA baseline (Figure 4).

Different regularization mechanisms produce mixed effects on the task-level accuracy improvement. However, none of these standard regularizers match the aggregate gains from adaptive merging (0.672). Adaptive merging procedure appears to provide a more effective form of regularization, potentially through the over-parameterized coefficient space or implicit ensembling, that standard techniques do not capture. That said, the comparable performance between

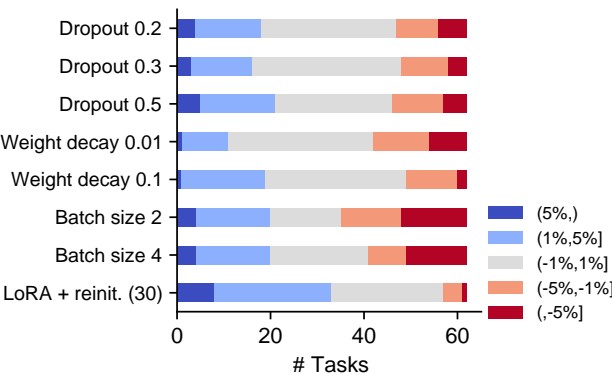

*(a)* % acc. improvement compared to LoRA baseline, each regularization technique applied to target-task LoRA training.

| Method | Avg. accuracy |
|---|---|
| Dropout 0.2 | 0.655 |
| Dropout 0.3 | 0.656 |
| Dropout 0.5 | 0.659 |
| Weight decay 0.01 | 0.647 |
| Weight decay 0.1 | 0.657 |
| Batch size 2 | 0.626 |
| Batch size 4 | 0.644 |
| LoRA + reinit. (30) | 0.672 |
| LoRA | 0.657 |

*(b)* Avg. downstream task accuracy for each regularization method. "LoRA" indicates the baseline target-task LoRA performance.

*Figure 4.* Performance of target-task LoRAs trained with regularization technique, compared with target-task LoRA merged with randomly reinitialized LoRAs ("LoRA + reinit. (30)")

reinitialized LoRAs and the curated ones (Figure 3) still suggests the benefit of adaptive merging is closer to a structural regularization effect than genuine cross-task transfer.

## Q4. Why does including target LoRA in the pool reduce the performance gap among selection methods?

To better understand why the choice of selection method matter less once the target LoRA is in the merging pool, we analyze how the distribution of learned merging coefficients shifts when the target-task LoRA is included. We focus on 20 LoRAs with the highest evaluation scores for the given target task. First, we normalize the learned merging coefficients to [0,1], then divide each by the sum of all 20 coefficients to convert them into percentage assigned to each LoRA. We then average these percentages across all target modules for each downstream task and plot their distribution in Figure 5, highlighting the percentage assigned to the best-performing recycled LoRA in the pool (Figure 5a, orange) — i.e., the recycled LoRA with the highest evaluation score for the given target task — and the percentage assigned to the target-task LoRA when it is included (Figure 5b, orange).

As seen in Figure 5, target-task LoRA receives a substan-

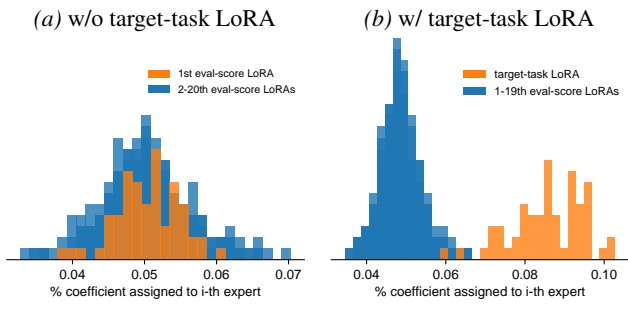

*Figure 5.* Distribution of merging coefficients across 20 evaluation-based LoRAs for 62 downstream tasks, a) without and b) with the target-task LoRA in the merging pool.

tially higher coefficient than the other 19 LoRAs (Figure 5b), whereas the recycled LoRA with the highest evaluation score receive similar coefficients as other LoRAs despite achieving the highest performance on the target task (Figure 5a). In Section H, we further show that the distribution pattern is more varied when target-task LoRA is excluded, LoRAs with lower evaluation scores sometimes receiving higher values than others on certain tasks. Our result suggests that the coefficient optimization discovers the target-task LoRA as having the most relevant knowledge for the task, causing other LoRAs to be overshadowed.

## Q5. Do controlled evaluations overstate adaptive merging gains?

While we obtain weak results from recycled LoRA, previous adaptive merging works have established substantial gains, and more broadly speaking, transfer learning has a long history of success. How to reconcile this discrepancy? To investigate, we apply the same recycling method as our main method in §3, but replace the recycled LoRA pool with all the target-task LoRAs trained on 100 examples ("in-house" LoRAs). We evaluate the accuracy of the 62 downstream task LoRAs on each other's training and validation examples to use evaluation-based selection. For each task, we combine the target-task LoRA with the top-$(k-1)$ ranking LoRAs from other tasks, varying $k \in \{5, 10, 20, 30\}$. Figure 6a shows using these "in-house" LoRAs leads to consistently stronger results than that of recycled LoRAs, with mild growth from higher $k$ values. This demonstrates that positive transfer is possible when recycling LoRAs trained in a controlled setting on tasks more relevant to the target.

Given this finding, it is perhaps surprising that the nearly 1,000 LoRAs in our recycled LoRA pool are seldom helpful for our set of 62 downstream tasks. Is the region where positive transfer occurs so narrow that it rarely arises naturally? To investigate, we fix the number of merged LoRAs $k = 10$ and intentionally omit the top-$m$ LoRAs from other target tasks, where $m \in \{5, 10, 15, 20\}$. This acts as a sliding

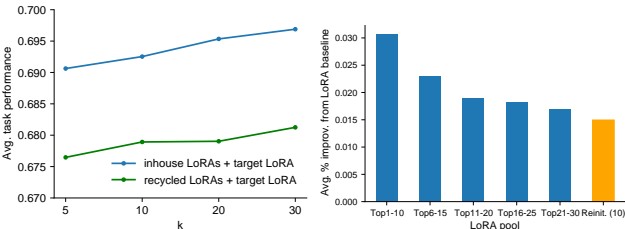

*(a) In-house vs recycled LoRAs. (b) Effect excluding top LoRAs.*

*Figure 6.* (a) In-house LoRAs outperform recycled LoRAs. (b) Excluding top-ranked in-house LoRAs degrades merging performance, converging to the randomly reinitialized LoRA performance.

window of fixed size. From Figure 6b, as we omit more top-ranked LoRAs, we observe clear performance deterioration, all the way till it reaches almost the same performance as LoRAs with random parameters. This suggests that positive transfer decays rapidly with task dissimilarity, i.e., transfer relies on highly relevant source LoRAs that others cannot substitute. Despite the fact that our recycled LoRA pool is dramatically larger than those considered in past work, it may still fall below the coverage needed to provide reliable gains in practice, or else demand a categorically different approach to discovering rare matches.

## 6 Related works

**Model Recycling**   Besides the merging methods studied in our work, several works have explored using specialized LoRA modules to improve performance on a target task, particularly when only a limited number of examples are available. AdapterFusion (Pfeiffer et al., 2021) addresses this by learning an attention-based fusion module (Vaswani et al., 2017) to combine task-specific LoRAs (Rebuffi et al., 2017; Houlsby et al., 2019) on a target dataset. Within each layer, the fusion module uses the input representation of the pre-trained model as a query, while the key and value are derived from the adapters' output representations. Other routing-based methods propose a mixture-of-Experts approach to dynamically select and aggregate outputs from multiple LoRA modules, rather than averaging their parameters; MeteoRA (Xu et al., 2025) uses a learned gating mechanism to activate a predetermined number of LoRAs for each token; similarly, MoLE (Wu et al., 2024) employs layer-wise routers to reweight and aggregate outputs from LoRA modules and uses a load-balancing loss to ensure all modules are utilized. We refer the reader to the comprehensive survey on model MoErging (Yadav et al., 2025b) for a complete overview and taxonomy of relevant methods. (Kahana et al., 2026) explores discovering "hidden gems" among recycled models available on Hugging Face and propose a method to identify them efficiently.

**Model Merging**   Model merging (Matena & Raffel, 2022; Choshen et al., 2022) aims to merge independently trained models with identical architecture into one model that maintains their combined capabilities. Most methods rely on some form of parameter-space aggregation (Utans, 1996; McMahan et al., 2017) to interpolate the constituent models. This is effective because when models share the same initialization, as in transfer learning, their parameters reside in a linearly connected low-loss region (Frankle et al., 2020; Altıntaş et al., 2025). In the context of multiple expert models fine-tuned on different tasks, Ilharco et al. (2023b) introduced task vectors, defined as the parameter change from a pre-trained model to its fine-tuned version. They showed that these vectors can be combined to create a single generalist model.

## 7 Conclusion and Takeaways

We conducted the first large-scale evaluation of adaptive merging using nearly 1,000 user-contributed LoRAs from Hugging Face, assessing whether these methods can recycle public LoRAs to improve target-task performance in realistic conditions.

Our findings paint a nuanced picture. While adaptive merging can clearly improve over the base model, the benefits largely disappear when compared against a simple baseline: training a LoRA directly on the target-task data. More strikingly, when the target-task LoRA is included in the merging pool, the choice of which LoRAs to recycle has minimal impact—even randomly initialized LoRAs perform comparably to carefully selected ones. We only find "in-house" LoRAs to outperform this randomly initialized LoRAs when merging with target-task LoRA. This suggests that reported successes in prior work may depend critically on close task connection that's easily available in controlled experimental conditions but rare in practice.

Two interpretations are consistent with these results: either 1) a different selection method is needed to maximize positive transfer from hub LoRAs, or 2) recycled LoRAs are generally incompatible with target-task LoRAs due to differences in training data, hyperparameters, or other factors, though isolating these causes are challenging given sparse metadata for publicly available LoRAs.

Taken together, these results carry practical implications: researchers considering adaptive merging should first verify that it outperforms a target-task LoRA baseline and embrace in-the-wild evaluation settings. More broadly, our findings highlight a gap between the promise of LoRA recycling and its current practical utility, pointing to the need for reconsidering research methodology—moving beyond controlled benchmarks to realistic heterogeneous pools—and establishing better infrastructure around public LoRA repositories, standardized metadata, licensing, and training provenance.

## Acknowledgement

Resources used in preparing this research were provided, in part, by the Province of Ontario, the Government of Canada through CIFAR, the Digital Research Alliance of Canada, and companies sponsoring the Vector Institute. We also thank Mohammed Muqeeth and John Mason for helpful discussion during the project and feedback on the draft.

## Impact Statement

Our work contributes to a realistic assessment of adaptive merging methods for recycling publicly available LoRAs. Although adaptive merging has been promoted as a way to repurpose the growing body of publicly released LoRA adapters, our findings caution that these methods evaluated in realistic, heterogeneous settings face several challenges. Limited documentation in public LoRA repositories introduce additional difficulties in choosing which adapters to recycle. That said, merging recycled LoRAs can still benefit from improved selection or merging methods designed for in-the-wild conditions. We hope these results encourage, rather than discourage, further work on adaptive merging, evaluated under realistic settings.

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

# A   Recycled LoRA Analysis

We analyze the distribution of the recycled LoRA's ranks, target modules, and sources and include the result in Figure 7.

*(a)* # of recycled LoRAs uploaded per contributor.

| Percentile | # recycled LoRAs |
|------------|------------------|
| Avg. | 3.259 |
| 25% | 1 |
| 50% | 1 |
| 75% | 2 |
| 90% | 6 |
| 95% | 9 |
| 99% | 25.66 |
| max | 99 |

*(b)* Rank distribution of recycled LoRAs

| Rank | # recycled LoRAs |
|------|------------------|
| 16 | 286 |
| 8 | 274 |
| 64 | 200 |
| 32 | 137 |
| 256 | 29 |
| 128 | 24 |
| 96 | 5 |
| 4 | 1 |
| 25 | 1 |
| 512 | 1 |

*(c)* Rank distribution of recycled LoRAs grouped by target module types.

| Combination of target module | # recycled LoRAs |
|------------------------------|------------------|
| **Complete FFW + Attn** | |
| FFW (Down, Gate, Up) + Attn (K, O, Q, V) | 766 |
| FFW (Down, Gate, Up) + Attn (K, O, Q, V) + LM_Head | 39 |
| FFW (Down, Gate, Up) + Attn (K, O, Q, V) + Embed + LM_Head | 2 |
| FFW (Down, Gate, Up) + Attn (K, Q, V) | 1 |
| **Attention Only** | |
| Attn (Q, V) | 91 |
| Attn (K, Q, V) | 22 |
| Attn (K, O, Q, V) | 14 |
| Attn (O, Q, V) | 4 |
| Attn (Q) | 4 |
| L30.K + L30.V + L31.K + L31.V | 1 |
| **FFW Only** | |
| FFW (Down, Gate, Up) | 1 |
| **Mixed FFW + Attn** | |
| FFW (Gate) + Attn (K, O, Q, V) | 10 |
| FFW (Gate, Up) + Attn (K, O, Q, V) | 2 |
| FFW (FC_In, FC_Out) + Attn (K, Q, V) + Out + WTE | 1 |

*Figure 7.* Recycled LoRA summary statistics.

# B   Downstream Task Details

## B.1   Recycled LoRA evaluated on downstream tasks

We evaluate the recycled LoRAs on 100 samples from each of the 62 downstream task and plot the distribution of their evaluation scores (Figure 8). The computed scores are used for the evaluation-based selection.

Furthermore, we observe substantial variation in the concentration of relevant LoRAs across tasks (Figure 9). In our experiments in §4, we rescale the target modules of 958 recycled LoRAs to measure positive transferability across the 62 downstream tasks. For tasks on the far left of Figure 9, none of the LoRAs achieve notable improvement from prompting. On the other hand, tasks in the far-right of Figure 9 have a mix of highly relevant LoRAs that achieve substantial % improvement from prompting alongside irrelevant ones with little to negative benefit. This finding highlights that the utility of recycled LoRAs should be assessed across downstream tasks spanning multiple domains, as their relevance may depend heavily on task selection.

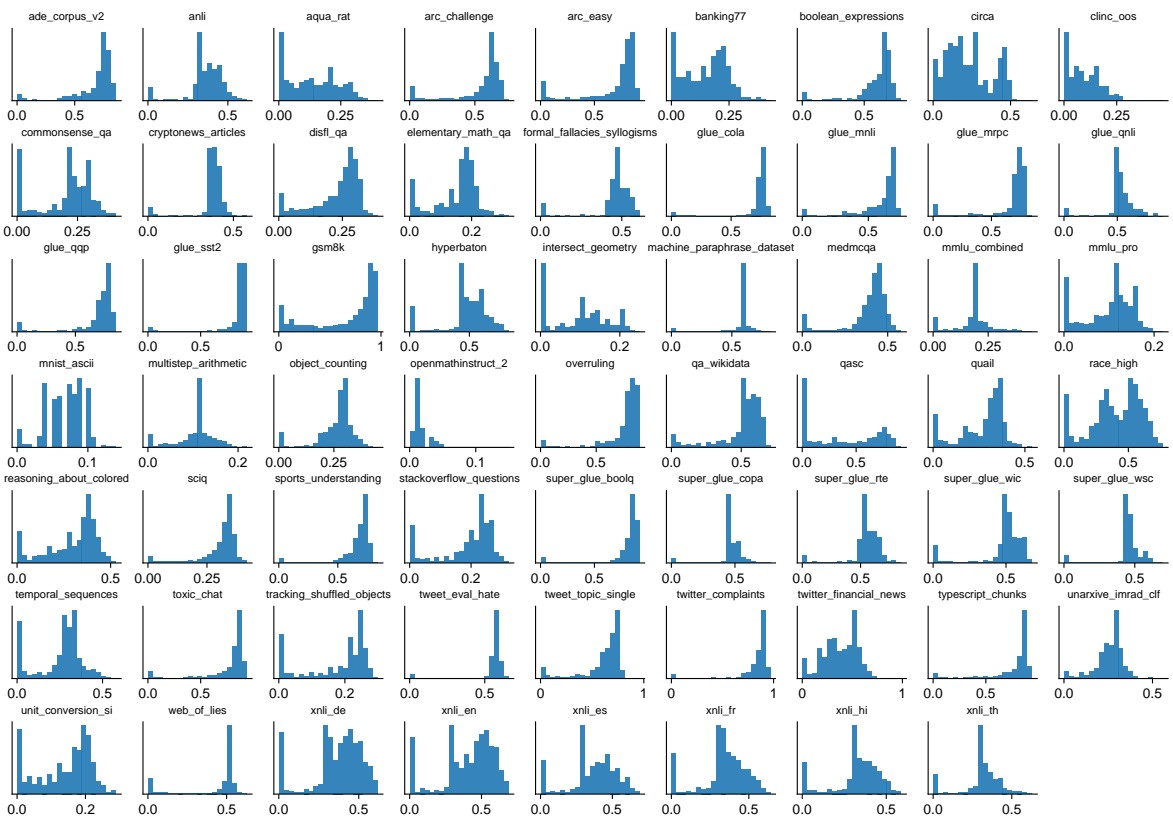

*Figure 8.* Distribution of recycled LoRA evaluation scores (100 data samples) across 62 tasks.

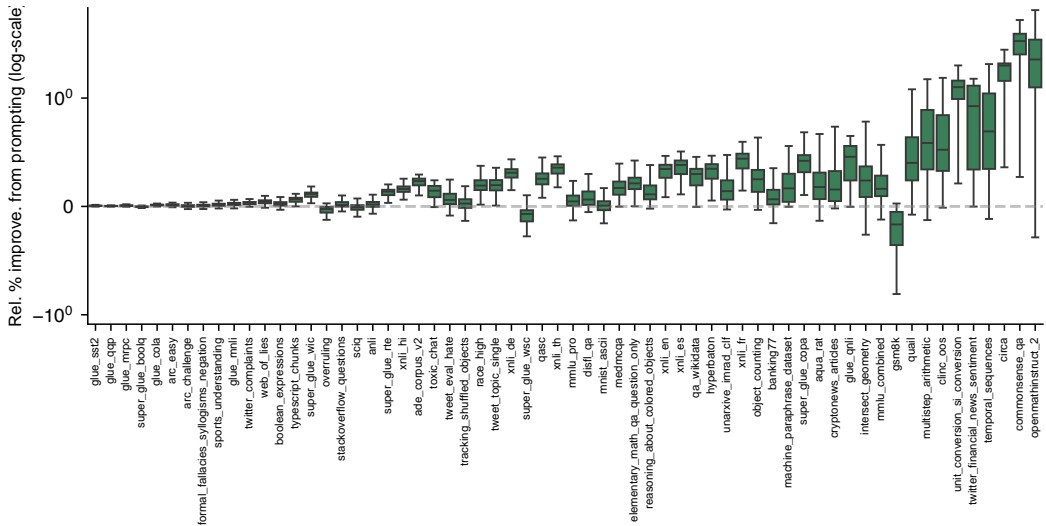

*Figure 9.* Distribution of relative % improvement from prompting by task. Most tasks have many LoRAs to gain relative % improvement from, some only with a few relevant LoRAs that can transfer positive knowledge, and a handful of tasks with little or worse performance gain after further optimization.

### B.2  Downstream task description

In addition to the downstream tasks included in SuperGLUE (Wang et al., 2019a), GLUE (Wang et al., 2019b), BIG-Bench (Srivastava et al., 2023), and the curated tasks from (Je & Raffel, 2025), we also include the following tasks in our

experiments:

**Math and science.** AQuA-RAT (Ling et al., 2017) consists of algebraic word problems with multiple-choice answers and rationales. GSM8k (Cobbe et al., 2021) is grade-school math word problems that requires multi-step reasoning. OpenMathInstruct-2 (Toshniwal et al., 2024) is a math instruction-tuning dataset with synthetically generated problem-solution pairs. ARC-Easy (arc_easy) and ARC-Challenge (arc_challenge) (Clark et al., 2018) are subsets of the ARC science, which tests scientific knowledge requiring logical reasoning and multi-fact inference. QASC (Khot et al., 2020) is a grade-school science question-and-answer dataset that requires combining multiple factual statements. SciQ (Welbl et al., 2017) contains crowdsourced science exam questions spanning topics such as physics, chemistry, and biology.

**Document understanding.** unarxive_imrad_clf (Saier et al., 2023) involves classifying arXiv paper sections into IMRAD categories (Introduction, Methods, Results, and Discussion). machine_paraphrase_dataset (Wahle et al., 2022) focuses on detecting whether a given document extracted from online sources (Wikipedia, arXiv, and student theses) has been paraphrased using automated tools.

**Multilingual inference.** XNLI (Conneau et al., 2018) is a multilingual extension of MNLI (Williams et al., 2018), translated into 14 different languages. We take their English (xnli_en), German (xnli_de), Spanish (xnli_es), French (xnli_fr), Hindi (xnli_hi), and Thai (xnli_th) subsets.

**Domain-specific classification.** The stackoverflow_questions dataset [3] contains titles and bodies of StackOverflow posts, along with labels derived from document metadata such as view counts and the number of up-votes. The cryptonews_articles_with_price_momentum_labels dataset (in short, "cryptonews_articles") [4] contain cryptocurrency-related news articles, labeled by their binary impact on the price movements. The tweet_topic_single dataset (Antypas et al., 2022) is a multi-label topic classification of tweets. The twitter_complaints (Preoţiuc-Pietro et al., 2019) labels tweets by whether the content contains customer complaints.

## C  Training Details

All training is conducted on 1xl40 or 1xH100 GPU. Below we provide a summary table of # trainable parameters and compute used for each approach (Table 3). $O(E)$ denotes the LoRA size, $k$ the number of selected LoRAs in the pool, $K$ the number of all available LoRAs, $L$ the number of layers in the base model, $M$ the number of target modules. Model-level merging coefficient uses $k$ trainable parameters, whereas module-level merging coefficient requires $k \times L \times M$ trainable parameters. $\pi$-Tuning jointly tunes merging coefficients and the $k$ selected LoRAs, incurring additional $k \times O(E)$ parameters to train. Methods using non-random selection method necessitate evaluating all $K$ LoRAs in the pool on the downstream task data points (ours) or against the target-task LoRA ($\pi$-Tuning), which we denote as "K evals" in Table 3.

*Table 3.* Compute and parameter requirements for merging methods and target-task LoRA.

| Method | Trainable params | LoRA selection | Tuning steps |
|---|---|---|---|
| Simple averaging | 0 | O(1) | 0 |
| TSV | 0 | O(1) | 0 |
| TIES | 0 | O(1) | 0 |
| LoraHub | $k$ | O(1) | 100 |
| AdaMerging | $k \times L \times M$ | O(1) | 100 |
| $\pi$-tuning | $k \times L \times M + k \times O(E)$ | $K$ evals | 100 |
| Ours | $k \times L \times M$ | $K$ evals | 100 |
| Target-task LoRA | $O(E)$ | N/A | 100 |

### C.1  Merging Coefficient Training

For all adaptive merging methods that require training merging coefficients, we use a fixed set of 100 samples per target task. Below, we describe the training procedure for each method's merging coefficients.

**Ours** We train per-module merging coefficients on 80 training examples for 100 steps, with a learning rate of 5e-2, selecting the coefficients that achieve the lowest validation loss on the 20 validation examples. For our merging design space

---

[3] https://huggingface.co/datasets/pacovaldez/stackoverflow-questions
[4] https://huggingface.co/datasets/SahandNZ/cryptonews-articles-with-price-momentum-labels

exploration, we maintain this training procedure but vary the granularity of merging coefficients across per-module, per-layer, per-sublayer, and per-model levels.

**AdaMerging** We train per-module merging coefficients on 80 training examples for 100 steps, with a learning rate of 5e-2, selecting the coefficients that achieve the lowest validation loss on the 20 validation examples. AdaMerging uses linear activation with weights initialized to $\frac{1}{k}$.

We choose AdaMerging rather than its TIES-enhanced variant AdaMerging++ for two reasons. First, TIES underperforms simple averaging in our experiments. Second, applying AdaMerging++ to LoRAs would require around $k$ times the memory of a base model, which is computationally infeasible.

**$\pi$-Tuning** Similarly, we train per-module merging coefficients on 80 training examples for 100 steps, selecting the coefficients that achieve the lowest validation loss on the 20 validation examples. $\pi$-Tuning uses softmax activation with weights initialized to 0. $\pi$-Tuning jointly tunes merging coefficients and the selected LoRAs, requiring additional $k \times O(E)$ trainable parameters, on top of $k \times L \times M$ merging coefficients. As $\pi$-Tuning has the highest number of parameters among other methods, we choose lower learning rates of 1e-4 and 5e-5, and report on the average test split performance achieved by the two runs.

**LoraHub** Unlike other merging methods, LoraHub uses gradient-free optimization at the *per-model* level. As LoraHub uses model-level merging coefficient granularity, it only tunes $k$ parameters. Following the original LoraHub implementation, we use gradient-free optimization with the merging coefficients initialized to 0. After 100 optimization steps, we take the merging coefficients that achieve the lowest loss on the full 100 examples.

### C.2 Target-task LoRA Training

We train a target-task LoRA for each of the 62 downstream tasks using 100 data samples per task. We use an 80:20 split for training and validation. We train a rank-64 LoRA for 400 steps with a learning rate of 3e-4, LoRA initialized on feed-forward modules (down projection, gate projection, and up projection) and attention modules (K, O, Q, V). We choose the best checkpoint based on the validation loss.

## D  Merging Method Adjustment

**TIES Merging** To apply TIES to the selected 30 random LoRAs, we first convert each LoRA's A and B matrices into a full task vector B@A. This conversion is necessary because the ranks of A and B matrices vary among recycled LoRAs, and TIES requires parameter magnitude comparison among matrices of the same shape. To identify the optimal hyperparameter for TIES, we perform an extensive sweep over the percentage of weights pruned from each recycled LoRA, $p \in \{0.2, 0.4, 0.6\}$, and the coefficients assigned for each recycled LoRA, $C \in \{1, 0.3, \frac{1}{k}\}$. We report the average performance for each hyperparameter combination for 30 random LoRAs sampled using three seeds (Table 4). We find no clear winning configuration. Surprisingly, when the target-task LoRA is included in the pool, assigning a high coefficient degrades performance. TIES merging is performed separately for each target module for every layer.

*Table 4.* Avg. downstream task performance across TIES hyperparameter

| | LoRA coefficient | | | | | |
| | w/o target-task LoRA | | | w/ target-task LoRA | | |
| prune% | 0.3 | $\frac{1}{k}$ | 1 | 0.3 | $\frac{1}{k}$ | 1 |
|---|---|---|---|---|---|---|
| 0.2 | 0.47 | 0.47 | 0.29 | 0.44 | 0.47 | 0.06 |
| 0.4 | 0.47 | 0.47 | 0.46 | 0.45 | 0.46 | 0.16 |
| 0.6 | 0.47 | 0.47 | 0.49 | 0.47 | 0.46 | 0.26 |

**TSV Merging** Similarly as TIES, we perform TSV merging on the full task vector B@A, as the ranks of A and B matrices among recycled LoRAs vary. We select 30 random LoRAs, sampled across three seeds. TSV merging performs singular value decomposition (SVD) on each full matrix and extracts the top-8 singular vectors per LoRA. Each LoRA is assigned a weight of $\frac{1}{30}$ for its contribution to the shared full matrix computed by TSV. TSV merging is performed separately for each target module for every layer.

**LoraHub** The original LoraHub implementation assumes that all LoRA A and B matrices have the same rank, allowing direct summation of A and B matrices across LoRAs in the pool to produce a single summed matrix. Since the recycled

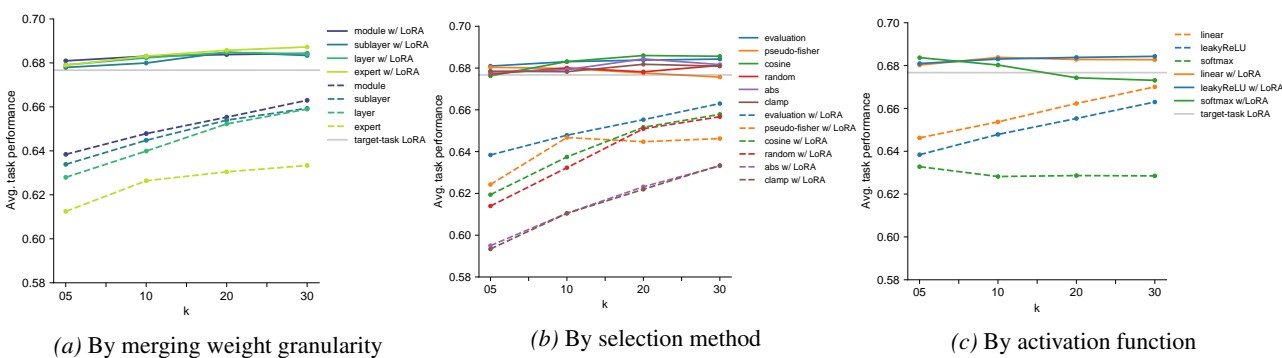

*(a)* By merging weight granularity      *(b)* By selection method      *(c)* By activation function

*Figure 10.* Average downstream task performance across 20 downstream tasks used in ablation, by the design choice in a) merging weight granularity, b) selection method, and c) activation.

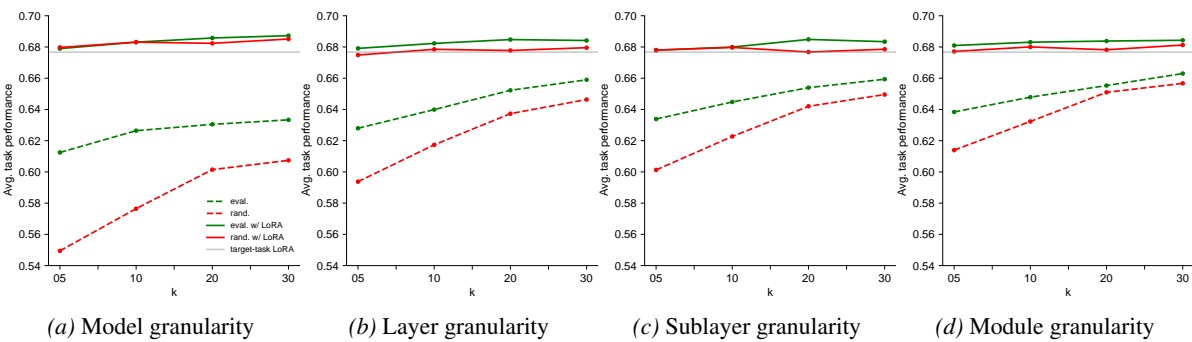

*(a)* Model granularity    *(b)* Layer granularity    *(c)* Sublayer granularity    *(d)* Module granularity

*Figure 11.* Performance gap between random and evaluation-based selections, without and with target-task LoRA in the pool. Though there is a notable gap between random and evaluation-based selections, including the target-task LoRA in the pool, closes this gap across all granularities.

LoRAs have varying ranks, we pad each LoRA to the max rank present in the pool before merging.

**$\pi$-Tuning** Since we do not have access to the training data of the recycled LoRAs, we compute a "Quasi-FIM", an approximation of Fisher Information Matrix (FIM), using the squared weights of the recycled LoRAs. To select the LoRAs based on this metric, we compute the cosine similarity between the quasi-FIM of the target-task LoRA and of each recycled LoRA, then select the recycled LoRAs with the highest cosine similarities.

# E Merging Design Space Exploration

**Impact of merging coefficient granularity, selection, and activation function.** Using our method, we run ablation studies on a set of 20 tasks to evaluate factors impacting merging performance, such as the granularity of merging coefficients and LoRA selection strategy. In Figure 12, we report the average % improvement across the selected ablation tasks over the prompting and LoRA baselines, both in settings without and with the target-task LoRA in the pool, with the number of experts $k = 30$ LoRAs in the pool. For the selection method comparison, we fix the merging coefficient granularity to module-level. We initialize the merging weight to 0. Figure 10 shows the average performance across the selected ablation tasks by merging weight granularity (Figure 10a), selection method (Figure 10b), and activation function (Figure 10c) with varying $k \in \{5, 10, 20, 30\}$, without (dotted line) and with (solid line) the target-task LoRA in the pool.

We find that the more granular merging coefficient and the evaluation-based selection lead to higher average % improvement and average downstream task performance when the target-task LoRA is excluded from the pool. Interestingly, the gap between evaluation and random selections is wider for coarser merging weight granularity (Figure 11). Leaky ReLU and linear activations yield a more stable increase in performance with varying $k$ compared to softmax activation. However, once the target-task LoRA is included, the impact of specific design choices becomes only marginal.

*(a)* Performance improvement compared against *prompting* across different design choices.

| Design choice | Method | w/o target-task LoRA | | w/ target-task LoRA | |
|---|---|---|---|---|---|
| | | avg. % diff | # outperformed | avg. % diff | # outperformed |
| Granularity | Model (k) | 0.138 | 19 | 0.192 | 20 |
| | Layer (k * L) | 0.163 | 19 | 0.188 | 19 |
| | Sublayer (k * L * 2) | 0.164 | 20 | 0.188 | 19 |
| | **Module** ( k * L * M ) | 0.167 | 19 | 0.189 | 19 |
| Selection | Abs | 0.137 | 18 | 0.186 | 20 |
| | Clamp | 0.138 | 19 | 0.185 | 19 |
| | Quasi-FiM | 0.151 | 19 | 0.180 | 19 |
| | Random | 0.161 | 19 | 0.186 | 19 |
| | Cosine | 0.162 | 19 | 0.190 | 19 |
| | **Evaluation** | 0.167 | 19 | 0.189 | 19 |
| Activation | Softmax | 0.136 | 15 | 0.177 | 20 |
| | Linear | 0.174 | 20 | 0.187 | 20 |
| | **Leaky ReLU** | 0.167 | 19 | 0.189 | 19 |

*(b)* Performance improvement compared against *target-task LoRA* across different design choices.

| Design choice | Method | w/o target-task LoRA | | w/ target-task LoRA | |
|---|---|---|---|---|---|
| | | avg. % diff | # outperformed | avg. % diff | # outperformed |
| Granularity | Model (k) | -0.043 | 6 | 0.011 | 15 |
| | Layer (k * L) | -0.018 | 7 | 0.007 | 14 |
| | Sublayer (k * L * 2) | -0.017 | 6 | 0.007 | 14 |
| | **Module** ( k * L * M ) | -0.014 | 6 | 0.008 | 15 |
| Selection | Abs | -0.044 | 5 | 0.005 | 13 |
| | Clamp | -0.043 | 4 | 0.004 | 13 |
| | Quasi-FiM | -0.030 | 4 | -0.001 | 9 |
| | Random | -0.020 | 6 | 0.005 | 15 |
| | Cosine | -0.019 | 8 | 0.009 | 15 |
| | **Evaluation** | -0.014 | 6 | 0.008 | 15 |
| Activation | Softmax | -0.056 | 4 | -0.004 | 10 |
| | Linear | -0.007 | 9 | 0.006 | 15 |
| | **Leaky ReLU** | -0.014 | 6 | 0.008 | 15 |

*Figure 12.* Ablation studies varying merging coefficients, granularity, and selection strategy. $k$ denotes the number of LoRAs in the pool, $L$ the number of model layers, $M$ the number of target modules. We **bold** our chosen configuration used in our method ("Ours").

**Impact of gradient-free vs. gradient-based optimization.** Furthermore, we analyze the impact of using gradient-free versus gradient-based merging weight optimization. Note that LoraHub learns merging coefficients at model-level granularity using randomly selected LoRAs with *gradient-free* optimization. We compare LoraHub's performance on the 20 selected ablation tasks against an approach that learns the merging coefficients in the same setup with *gradient-based* optimization and leaky ReLU activation. As shown in Table 5, gradient-based optimization slightly outperforms gradient-free optimization in average downstream task performance across the 20 ablation tasks, both without and with the target-task LoRA in the pool. Notably, gradient-free optimization without the target-task LoRA exhibits a performance decline when increasing from 20 to 30 LoRAs in the pool, suggesting that the gradient-free approach may converge to a suboptimal solution within the given compute budget as the number of LoRAs to merge increases.

*Table 5.* Gradient-free vs. Gradient-based optimization, average downstream task performance across the 20 ablation tasks.

| | w/o target-task LoRA | | w/ target-task LoRA | |
|---|---|---|---|---|
| k | Grad-free | Grad-based | Grad-free | Grad-based |
| 5 | 0.542 | 0.549 | 0.680 | 0.680 |
| 10 | 0.565 | 0.576 | 0.678 | 0.683 |
| 20 | 0.595 | 0.601 | 0.679 | 0.682 |
| 30 | 0.590 | 0.607 | 0.682 | 0.685 |

*Table 6.* Average performance across 62 downstream tasks in two ablated settings, using 1) Qwen3-4B-Instruct-2507 as base model instead of Llama 3.1 8B-Instruct and b) 10 target-task samples for optimization instead of 100 samples.

*(a)* Qwen3-4B-Instruct-2507 as base model.

| | Average accuracy | |
|---|---|---|
| Method | w/o LoRA | w/ LoRA |
| TSV (30) | 0.578 | 0.586 |
| TIES (30) | 0.578 | 0.577 |
| Simple avg. (30) | 0.569 | 0.583 |
| Arrow (30) | 0.204 | 0.257 |
| Adamerging (30) | 0.583 | 0.639 |
| $\pi$-Tuning (20) | 0.253 | 0.258 |
| LoraHub (30) | 0.542 | 0.636 |
| Ours (30) | 0.657 | 0.681 |
| Prompting (zero-shot) | 0.578 | |
| LoRA | 0.668 | |

*(b)* 10 data-budget setting.

| | Average accuracy | |
|---|---|---|
| Method | w/o LoRA | w/ LoRA |
| Adamerging (30) | 0.558 | 0.557 |
| $\pi$-Tuning (20) | 0.546 | 0.549 |
| LoraHub (30) | 0.517 | 0.587 |
| Ours (30) | 0.578 | 0.566 |
| Prompting (zero-shot) | 0.578 | |
| LoRA | 0.546 | |

## F  Merging Method Evaluation Ablation

### F.1  Qwen model family

We fully replicate our experiments from §5 using Qwen3-4B-Instruct-2507 (Team, 2025) and report the performance across merging methods (Table 6a) and effectiveness of selection method without and with the target-task LoRA in the pool (Figure 13a).

**Performance across merging methods** Similar to our results using Llama 3.1 8B-Instruct from §5, we still find that 1) adaptive merging outperforms non-adaptive merging methods in settings both without and with the target-task LoRA, 2) the gap among methods is notably decreased once the fine-tuned LoRA is included, and 3) gradient-based tuning (Adamerge, Ours) leaves a gap over gradient-free selection (LoraHub) (Table 6a). "Ours" in this setup outperforms other adaptive merging methods considered. One notable difference is the decreased $\pi$-tuning performance. For $\pi$-tuning, we use the same learning rates (1e-4, 1e-5) and report the averaged performance as in the original Llama experiments, but observe a decreased overall performance. Upon investigation, we find that the "Quasi-FIM" selection method used for $\pi$-tuning LoRA selection (see Section D $\pi$-Tuning for detail) is biased toward hub LoRAs that make larger changes in the model, with the average $L^2$ norm of Quasi-FFIM LoRAs over two times as high as those of evaluation-based or random LoRAs. We suspect this leads to more severe interference. This highlights the practical challenge of finding task-relevant LoRAs in the wild using $\pi$-Tuning when the original training data for hub LoRAs is unavailable.

**LoRA selection method without and with target-task LoRA** We examine the impact of LoRA selection across varying numbers of LoRAs in the pool ($k$), without and with the target-task LoRA, following the setup in §5 (Figure 13a). Interestingly, we find that reinitialized LoRA ("Reinit.") performance has stronger performance than it did in Llama 3.1 8B-Instruct setting. We similarly find that the effect of selection decreases once the target-task LoRA is included, with the performance of reinitialized LoRAs and evaluation-based ("Eval.") LoRAs matching very closely.

### F.2  Lower data-budget setting for adaptive merging

We rerun our experiments in §5 only using 10 target-task data samples for adaptive merging methods, which require target-task data to tune the merging coefficients. At this lower data budget, we tune merging coefficients and target-task LoRAs for 100 steps, just as in the original tuning setup, without a validation set.

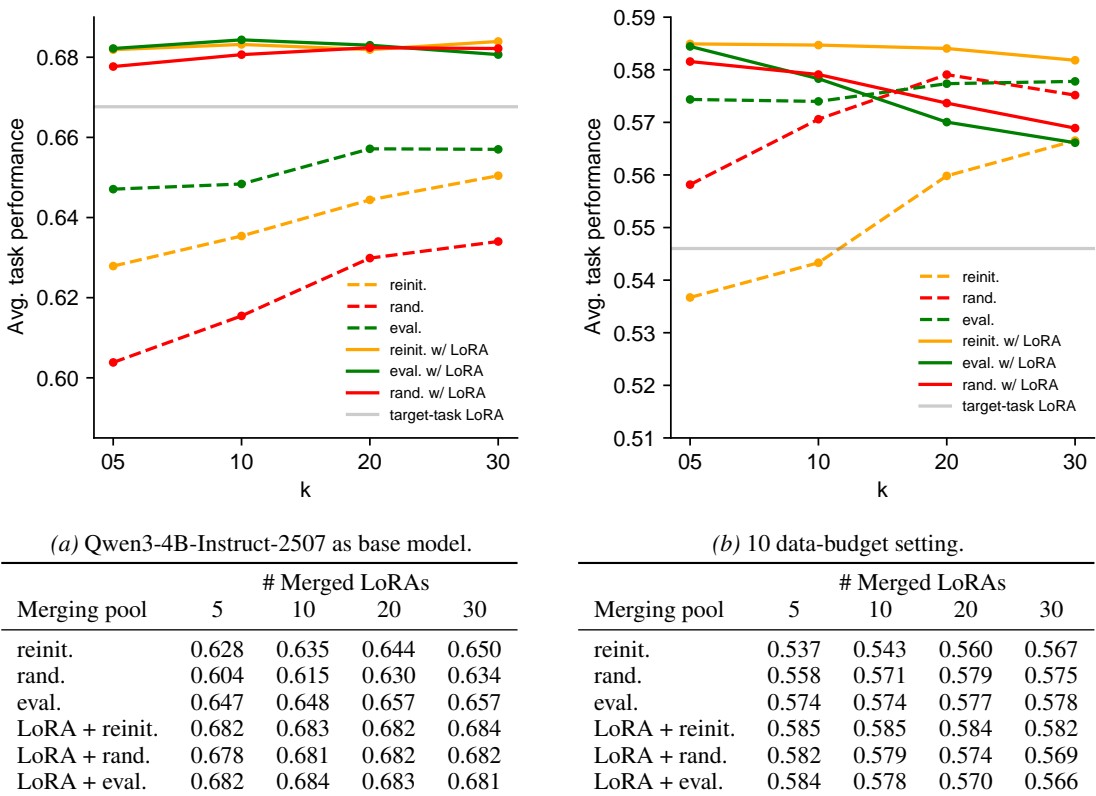

*(a)* Qwen3-4B-Instruct-2507 as base model.

| Merging pool | # Merged LoRAs | | | |
|---|---|---|---|---|
| | 5 | 10 | 20 | 30 |
| reinit. | 0.628 | 0.635 | 0.644 | 0.650 |
| rand. | 0.604 | 0.615 | 0.630 | 0.634 |
| eval. | 0.647 | 0.648 | 0.657 | 0.657 |
| LoRA + reinit. | 0.682 | 0.683 | 0.682 | 0.684 |
| LoRA + rand. | 0.678 | 0.681 | 0.682 | 0.682 |
| LoRA + eval. | 0.682 | 0.684 | 0.683 | 0.681 |

*(b)* 10 data-budget setting.

| Merging pool | # Merged LoRAs | | | |
|---|---|---|---|---|
| | 5 | 10 | 20 | 30 |
| reinit. | 0.537 | 0.543 | 0.560 | 0.567 |
| rand. | 0.558 | 0.571 | 0.579 | 0.575 |
| eval. | 0.574 | 0.574 | 0.577 | 0.578 |
| LoRA + reinit. | 0.585 | 0.585 | 0.584 | 0.582 |
| LoRA + rand. | 0.582 | 0.579 | 0.574 | 0.569 |
| LoRA + eval. | 0.584 | 0.578 | 0.570 | 0.566 |

*Figure 13.* Impact of LoRA selection in two ablated settings, using 1) Qwen3-4B-Instruct-2507 as base model instead of Llama 3.1 8B-Instruct and b) 10 target-task samples for optimization instead of 100 samples.

The 10-data setting introduces several brittlenesses. Evaluation-based selection is less robust, tuning is prone to overfitting, and target-task LoRA is weaker. Comparing across adaptive merging methods in the 10 data setting, we find that including target-task LoRA is not always consistently helpful as it has been in the 100 data setting (Table 6b). Using recycled hub LoRAs (i.e., "rand.", "eval.") still outperform randomly initialized LoRAs ("reinit."), but including the target-task LoRA in the pool no longer yields a consistent performance boost as the number of LoRAs in the pool ($k$) increases (Figure 13b).

### F.3 Different LoRA selection for non-adaptive merging

As non-adaptive merging methods are data-free, we default to random selection in the main experiment in §5. To isolate the merging method effectiveness from the impact of LoRA selection, we run experiments using evaluation-based LoRAs for non-adaptive merging methods (Table 7). "Random" refers to the original results using randomly selected LoRAs, "Eval" to the new results using evaluation-based LoRAs, "w/o" to the pool excluding target-task LoRA, "w/" to the pool including target-task LoRA. While evaluation-based LoRAs selection consistently improves upon the random selection, the non-adaptive merging methods still underperform adaptive merging methods (§5, Table 2).

*Table 7.* Avg. downstream task performance of non-adaptive merging methods using random LoRAs and evaluation-based LoRAs, without and with the target-task LoRA in the pool. Avg. prompting and LoRA baselines are included as reference.

| Method | Selection | | | |
|---|---|---|---|---|
| | Random (w/o) | Random (w/) | Eval. (w/o) | Eval. (w/) |
| TSV (30) | 0.488 | 0.522 | 0.504 | 0.502 |
| TIES (30) | 0.467 | 0.466 | 0.466 | 0.463 |
| Simple avg. (30) | 0.496 | 0.536 | 0.543 | 0.577 |
| Arrow (30) | 0.471 | 0.613 | 0.564 | 0.639 |
| Prompting (zero-shot) | 0.467 | | | |
| LoRA | 0.657 | | | |

# G    Merging Method Performance Across the Downstream Tasks

In this section, we show merging performance across the 62 downstream tasks by merging method (Figure 14) and by varying the number of LoRAs in the pool ($k$) for $\pi$-Tuning, LoraHub, and our method (Figure 15).

# H    Distribution of Merging Coefficients among LoRAs in the Pool

In addition to the merging coefficient distribution analysis across the 20 LoRAs in §5, we show the merging coefficient distributions within each downstream task (Figure 16). The most notable pattern is the skewed weight assignment to the target-task LoRA across all tasks (yellow bar, Figure 16b), whereas the weights are more or less evenly distributed across the 20 LoRAs when the target-task LoRA is not included in the pool (Figure 16b).

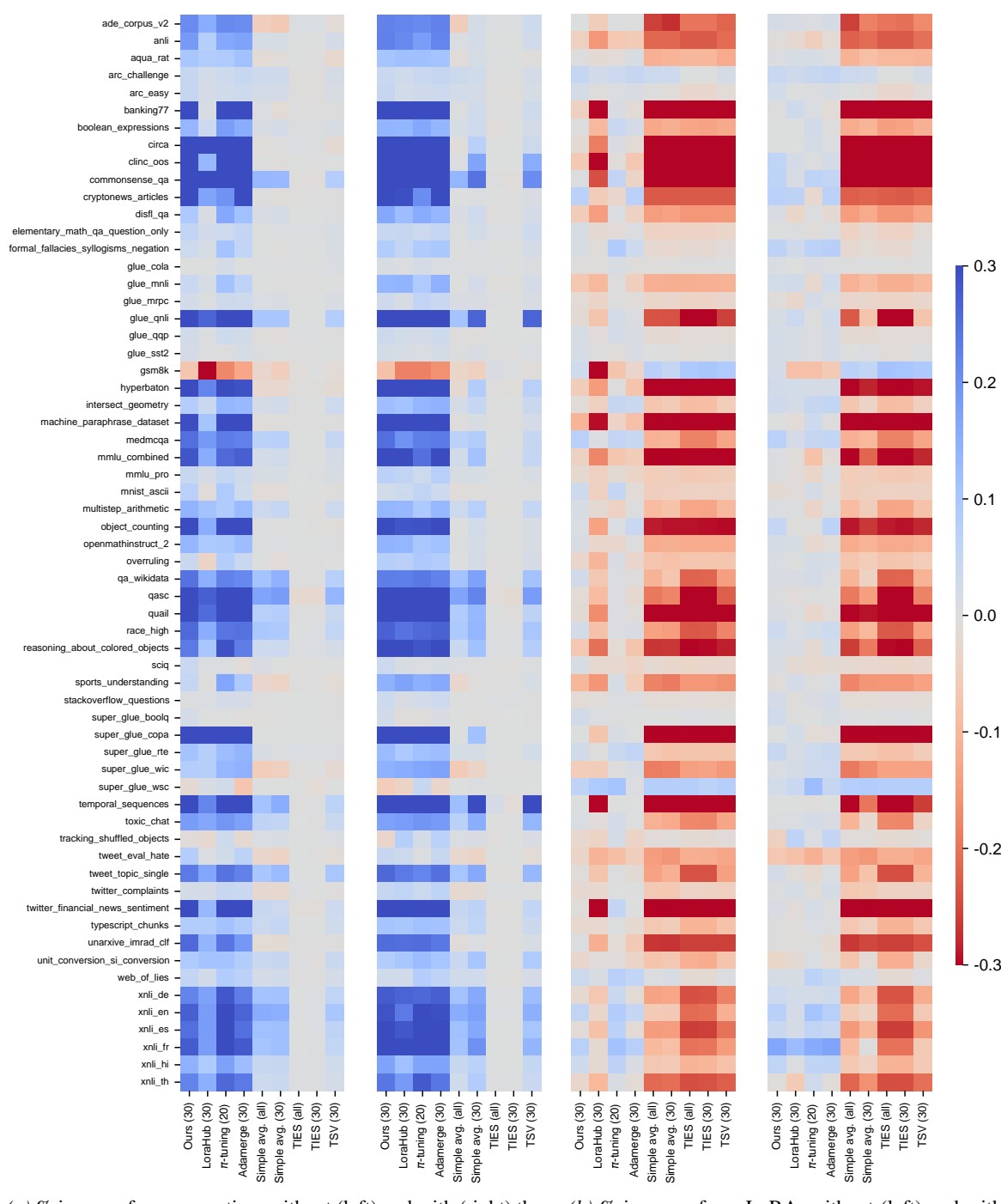

*(a) % improv. from prompting, without (left) and with (right) the target-task LoRA*

*(b) % improv. from LoRA, without (left) and with (right) the target-task LoRA*

*Figure 14.* % improvement from (a) prompting and (b) LoRA baselines, across the 62 downstream tasks for each merging method.

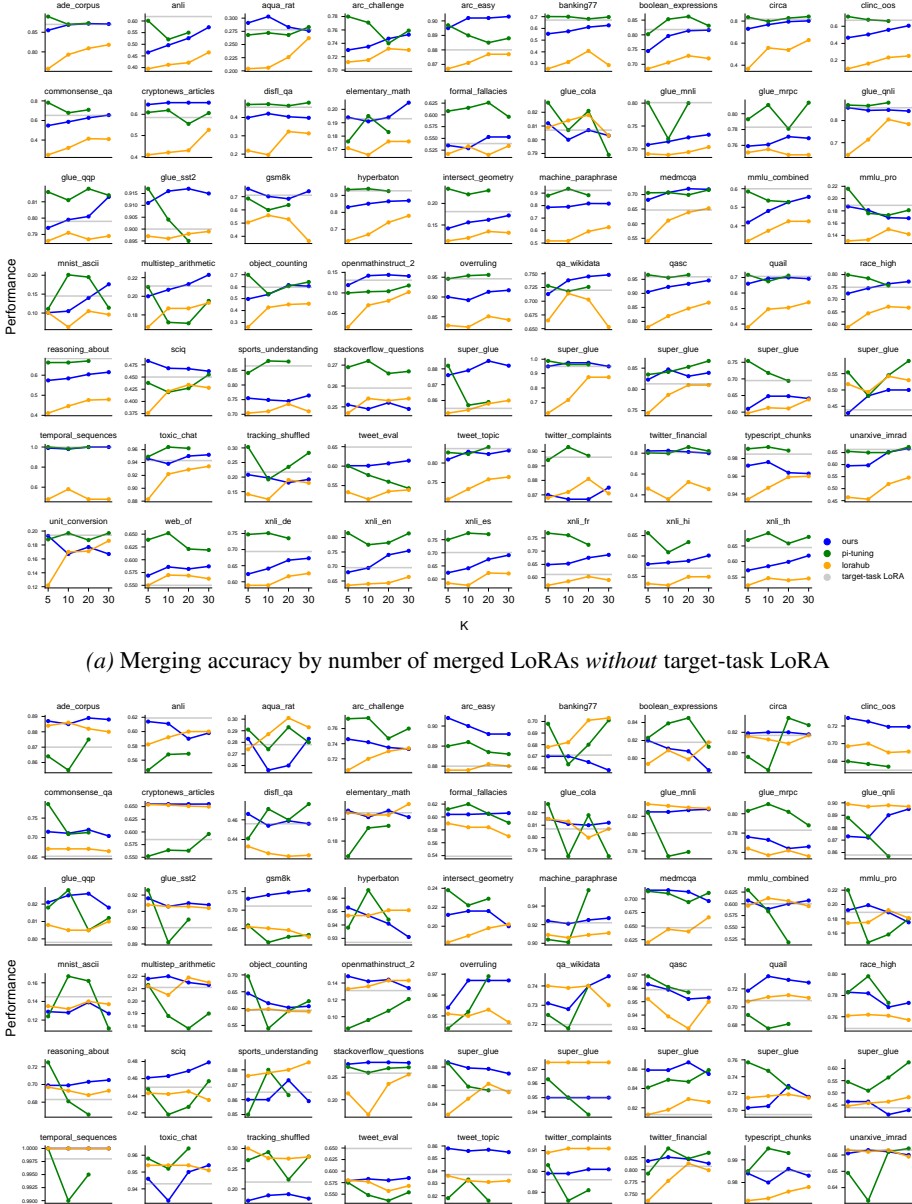

*(a)* Merging accuracy by number of merged LoRAs *without* target-task LoRA

*(b)* Merging accuracy by number of merged LoRAs *with* target-task LoRA

*Figure 15.* Merging accuracy by number of merged LoRAs without (a) and with (b) for $\pi$-Tuning, LoraHub, and our method.

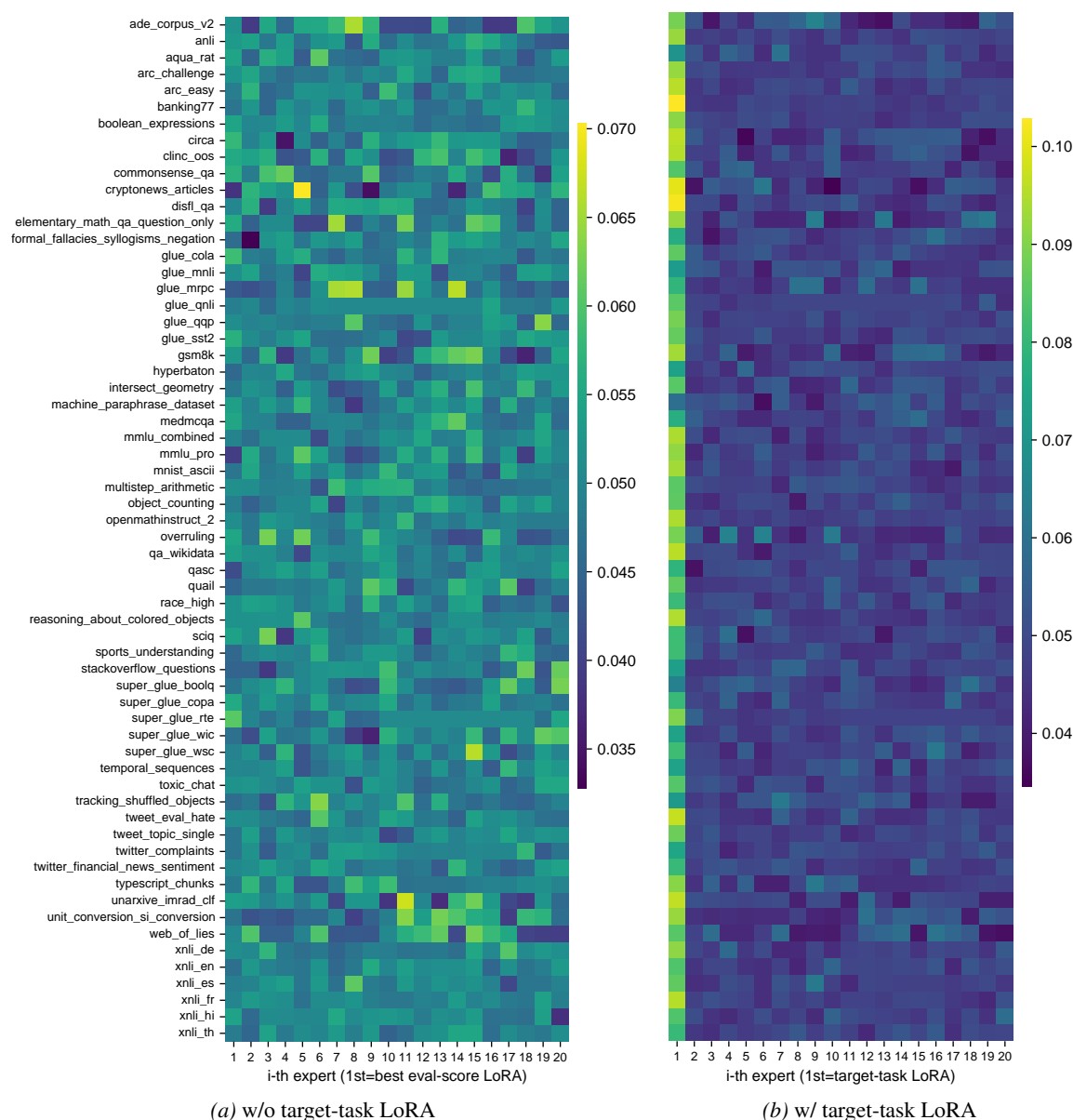

*(a) w/o target-task LoRA*  *(b) w/ target-task LoRA*

*Figure 16.* % coefficients assigned to 20 LoRAs in the pool across the 62 downstream tasks, in a setting a) without and b) with the target-task LoRA. In a), 1st expert has the highest evaluation score for the given downstream task. In b), 1st expert is the target-task LoRA, and the 2nd LoRA has the highest evaluation score for the given downstream task.

