# OpenReview forum: "The Appeal and Reality of Recycling LoRAs with Adaptive Merging"
_ICML.cc/2026/Conference — ICML 2026 regular_

### Official Review · Reviewer_ChpT · 2026-03-07

**Soundness:** 3
**Presentation:** 3
**Significance:** 3
**Originality:** 3
**Overall Recommendation:** 4
**Confidence:** 4

**Summary:**

The paper introduces a more realistic model merging setting, where LoRA experts are collected from Hugging Face rather than selected from a curated set of bespoke LoRAs. Based on this setting, the paper presents several findings about adaptive merging, including:

1. Adaptive merging methods usually require a small labeled dataset for merging; however, the resulting merged model often underperforms a model that is directly fine-tuned on the same small dataset.

2. The performance of random selection can be comparable with sophisticated model selection methods.

3. Adaptive merging may behave similarly to a regularization, as merging randomly sampled model weights achieves performance similar to merging fine-tuned weights.

Numerical experiments based on LLaMA-3.1-8B-Instruct are provided to support these claims.

**Compliance With Llm Reviewing Policy:**

Affirmed.

**Final Justification:**

All of my concerns has been solved, good paper.

**Key Questions For Authors:**

1. Since adaptive merging methods typically require a portion of target-task data for parameter tuning, and often perform poorly in the wild setting when the task-specific LoRA is unavailable, would it be more practical for users to directly fine-tune a model on the available task-specific data instead of using model merging?

2. Section 5.4 mentions that "Section 5.3 shows that although evaluation-based selection clearly outperforms random when the target-task LoRA is excluded, the performance gap becomes negligible when it is included." However, the numerical results referred to in Section 5.3 (Table 2 and Figure 2) do not explicitly highlight the comparison between random selection and evaluation-based selection. The only possible comparison appears to be between “Ours” and “AdaMerging” (if the activation difference is ignored), but their performance is already very close when the target-task LoRA is excluded. Could the authors clarify where this comparison is shown?

**Limitations:**

Referring to the weaknesses and questions above, the current work has several limitations. First, as the practical value of adaptive merging methods is limited due to their data-dependent nature, further research on discovering issues in adaptive merging may have limited significance. Second, the description of the proposed method and some experiment settings are not clear, and providing more details would improve the presentation. Finally, the numerical results do not show strong advantages of the proposed method over existing adaptive merging baselines. As a result, while the paper reveals some limitations of adaptive merging, it does not provide clear improvements over existing approaches.

**Strengths And Weaknesses:**

Strengths:

1. The idea of testing model merging methods "in the wild" is interesting. There are real world demanding that users would like to directly merging existing LoRAs from HuggingFace to satisfy the performance requirements across multiple tasks.

2. The paper provides several insights into the behavior of current merging methods when applied to LoRAs collected from Hugging Face.

3. The paper designed various experiments to support its claims.


Weaknesses:

1. The "Ours" method should be defined more clearly. The author should add more contents to further describe the proposed method. Besides, "Ours" methods does not show significant performance improvement comparing to other adaptive merging baselines in Table 2.

2. The current experiments are limited to LLaMA-3.1-8B-Instruct, evaluating on a wider range of backbone models would further support the claims.

3. The paper mainly discusses issues of adaptive merging in realistic applications. However, adaptive merging has been criticized for relying on task-specific samples to optimize coefficients, which contradicts the original motivation of data-free model merging. When such data is available, directly fine-tuning on the task data may be a simpler and stronger alternative. As a result, the practical value of adaptive merging methods may be limited, and further research on discovering issues in adaptive merging would be less significant.

---

> ### Author Rebuttal · Authors · 2026-03-31
>
> We appreciate your detailed comments and feedback. We’ve addressed each point below.
>
> ## 1. Ours method
>
> > "The "Ours" method should be defined more clearly. The author should add more contents to further describe the proposed method."
>
> The adaptive merging methods we consider specify a LoRA selection strategy, merging coefficient granularity, and tuning approach; and we sweep possible options in this design space to identify the best combination, yielding “Ours” (details in Section 3, summary). More concretely, given the original target module weight $W_0$, $k$ hub LoRAs for the target module each with low-rank $A_i$ and $B_i$ matrices, scaling coefficient $s \in R^{k}$, learnable merging coefficient $c \in R^{k}$, the merged output given the input $x$ is:
>
> $$ \tilde{c} = \text{LeakyReLU}(c) $$
>
> $$ \hat{W} = (s_1\tilde{c_1}A_1B_1 + s_2\tilde{c_2}A_2B_2 + … + s_k \tilde{c_k} A_kB_k) $$
>
> $$y = W_0(x) + \hat{W}(x)$$
>
> The coefficient granularity determines whether $c$ is distinct per module or shared across entire model/layers/sub-layers; the selection method changes the composition of A, B matrices; and the tuning method dictates whether $c$ is optimized via gradient-based or gradient-free methods. We can add these in the paper to make it clearer. Thank you for pointing it out and let us know if you have any other questions!
>
> > "..."Ours" methods does not show significant performance improvement comparing to other adaptive merging baselines in Table 2."
>
> We should clarify that the “Ours” method is not proposed to outperform other methods, but to thoroughly explore the adaptive merging design space and ensure that our reported results on merging performance are not subject to underexploration of the space. We will highlight this further in the paper.
>
> ## 2. Extension to different model family
>
> > "The current experiments are limited to LLaMA-3.1-8B-Instruct"
>
> This is a valid comment, also raised by reviewer tmrT. To fully address your concern, we have begun running our experiments end-to-end using the Qwen/Qwen3-4B-Instruct-2507 base model. The full replication is still running, as it involves scraping all available hub LoRAs (near 2000 for the new base model), running evaluation on the 62 downstream tasks, fine-tuning target-task LoRAs, and so on. While we believe our findings will hold across different base models, we aim to be able to share concrete results by the end of the discussion period!
>
> ## 3. Practicality of adaptive merging
>
> > "...the practical value of adaptive merging methods may be limited, and further research on discovering issues in adaptive merging would be less significant."
>
> This is an interesting point, but we respectfully disagree. We show that adaptive merging methods can outperform target LoRA trained on the same 100 data samples and data-free merging methods, making them appealing in practice. However, our work expands on this finding by evaluating them under a stricter, more realistic setup using LoRAs recycled "from the wild", since a commonly cited appeal of merging is that it enables recycling of in-the-wild LoRAs in data-restricted settings. Under this constraint, complex design choices such as LoRA selection or merging coefficient granularity yield surprisingly marginal returns, highlighting that such complex choices must be justified and tested in realistic conditions to fulfill the cited goal.
> Please let us know if we can clarify anything further or if you have any comments.
>
> > "would it be more practical for users to directly fine-tune a model on the available task-specific data instead of using model merging?"
>
> Directly fine-tuning on available data points is certainly the simplest approach. That said, based on our findings, 1) tuning merging coefficients over the target-task LoRA and randomly reinitializing LoRAs can yield further gain (Figure 3), and 2) merging with highly relevant LoRAs with the target-task LoRA is still effective (Section 5.5 shows that merging target-task LoRA with other highly relevant in-house trained LoRAs is still effective). In short, adaptive merging can still provide meaningful performance gains under certain conditions. We want to emphasize that our goal is not to discount prior work, but to identify when adaptive merging methods fall short and propose a realistic evaluation setup that surfaces these limitations.
>
> ## 4. Misc
>
> > "However, the numerical results referred to in Section 5.3 (Table 2 and Figure 2) do not explicitly highlight the comparison between random selection and evaluation-based selection."
>
> Thank you for pointing this out! This is a typo – the comparison between random and evaluation-based LoRA performance without and with the target-task LoRA is done in Figure 3, where “rand.” and “eval.” denote the LoRA pool without the target-task LoRA, and “LoRA + {selection}” denotes the pool with the target-task LoRA. We will fix the reference in the paper.

---

> > ### Author Rebuttal · Reviewer_ChpT · 2026-04-02
> >
> > I thank the authors for their rebuttal.
> >
> > The response looks great. I will update my score once the Qwen experiments are ready. I understand this takes time and effort.

---

> > > ### Author Response · Authors · 2026-04-05
> > >
> > > Thank you for your response!
> > >
> > > Our experiments using the Qwen baseline have finished running, and we report the two key results comparing merging methods (Table 2) and the effectiveness of the selection method (Figure 3). We sampled a total of 1956 LoRA modules from the Hugging Face hub for Qwen3-4B-Instruct-2507 after filtering (details in Section 4.1). Our experiment setup remains the same, with the same compute, data budget (80 for training, 20 for validation), and hyperparameters for merging weights tuning. For any methods using randomly selected LoRAs, we sample LoRAs using three random seeds and report the averaged accuracy.
> > >
> > > ## 1. Difference across merging methods
> > >
> > > | Method | w/o target | w/ target |
> > > |---|---|---|
> > > | TIES (30) | 0.578 | 0.577 |
> > > | TSV (30) | 0.578 | 0.586 |
> > > | Simple Avg (30) | 0.569 | 0.583 |
> > > | LoraHub (30) | 0.542 | 0.636 |
> > > | Adamerge (30) | 0.583 | 0.639 |
> > > | Pi-tuning (20) | 0.253 | 0.258 |
> > > | **Ours (30)** | 0.657 | 0.681 |
> > > *Prompting baseline*: 0.578
> > > *Fine-tune baseline*: 0.668
> > >
> > > **Findings** We still find that 1) adaptive merging outperforms non-adaptive merging methods in settings both without and with the target-task LoRA, 2) the gap among methods is notably decreased once the fine-tuned LoRA is included, and 3) gradient-based tuning (Adamerge, Ours) leaves a gap over gradient-free selection (LoraHub). "Ours" in this setup outperforms other adaptive merging methods considered.
> > >
> > > **Difference** One notable difference is the decreased $\pi$-tuning performance. For $\pi$-tuning, we use the same learning rates (1e-4, 1e-5) and report the averaged performance as in the original Llama experiments, but observe a decreased overall performance. Upon investigation, we find that the “Quasi-FIM” selection method (see Section 3 and Appendix C, $\pi$-Tuning for details) used for $\pi$-tuning LoRA selection is biased toward hub LoRAs that make larger changes in the model, with the average L2 norm of Quasi-FFIM LoRAs is 2+ times higher than those of evaluation-based or random LoRAs, which leads to more severe interference. Simple averaging using Quasi-FIM LoRAs also leads to poor performance. This highlights the practical challenge of finding task-relevant LoRAs in the wild using $\pi$-Tuning when the original training data for hub LoRAs is unavailable.
> > >
> > > ## 2. Reinit vs. Random vs. Eval. experts, w/o and w/ target-task LoRA
> > >
> > > As in Section 5.4, we examine the impact of LoRA selection across varying numbers of LoRAs in the pool ($k$), without and with the target-task LoRA in the pool. Interestingly, we find that reinitialized LoRA (“Reinit.”) performance in the Qwen3-4B-Instruct-2507 setting has stronger performance than it did for Llama 3.1 8B-Instruct setting. However, we still find that the effect of selection decreases once the target-task LoRA is included, with the performance of reinitialized LoRAs and evaluation-based (“Eval.”) LoRAs matching very closely.
> > >
> > > | Method | k=5 | k=10 | k=20 | k=30 |
> > > |---|---|---|---|---|
> > > | Reinit. | 0.628 | 0.635 | 0.644 | 0.650 |
> > > | Rand. | 0.604 | 0.615 | 0.630 | 0.634 |
> > > | Eval. | 0.647 | 0.648 | 0.657 | 0.657 |
> > > | | | | | |
> > > | LoRA+reinit. | 0.682 | 0.683 | 0.682 | 0.684 |
> > > | LoRA+rand. | 0.678 | 0.681 | 0.682 | 0.682 |
> > > | LoRA+eval. | 0.682 | 0.684 | 0.683 | 0.681 |
> > > *LoRA baseline*: 0.668
> > >
> > > We thank the reviewer for the suggestion and will add these findings to the paper. If we have addressed your concerns, we would appreciate it if you raised your score!

---

### Official Review · Reviewer_tmrT · 2026-03-13

**Soundness:** 3
**Presentation:** 3
**Significance:** 3
**Originality:** 2
**Overall Recommendation:** 5
**Confidence:** 4

**Summary:**

The authors investigate whether adaptive merging of LoRA adapters is actually useful in a realistic setting where the adapter pool comes from messy, heterogeneous public repositories rather than carefully curated in-house collections. The paper’s main empirical contribution is a large-scale study over 958 user-contributed LoRAs built on Llama 3.1 8B-Instruct, evaluated across 62 downstream tasks with both adaptive and non-adaptive merging methods. They find adaptive merging often improves over the base model, but usually does not deliver meaningful or consistent gains over simply training a target-task LoRA on the same small supervised dataset used to tune the merge. A second and more surprising finding is that, once the target-task LoRA is included in the merge pool, the identity of the other recycled LoRAs matters very little; even randomly reinitialized LoRAs perform similarly. The authors interpret this as evidence that much of the gain may come from regularization rather than meaningful cross-task transfer.

**Compliance With Llm Reviewing Policy:**

Affirmed.

**Key Questions For Authors:**

- The regularization hypothesis is central to the paper’s interpretation. Can you test it more directly? For example, how do results compare against other regularizers applied to the target-task LoRA training procedure, or against merging with structured noise of matched norm but different sparsity/rank profiles?

- How much of the conclusion depends on the base model family? Since all recycled LoRAs come from Llama 3.1 8B-Instruct variants, it would help to discuss whether the observed brittleness is expected to generalize to stronger bases, weaker bases, or models with different instruction tuning priors.

**Limitations:**

- The downstream setting is narrow in one important sense: every task gets the same small supervision budget. This is good for control, but it leaves open whether the conclusions hold in very low-shot or more data-rich conditions.

**Strengths And Weaknesses:**

- A major strength is that the paper asks the right question. The field has been drifting toward increasingly elaborate adapter-recycling methods, but many evaluations rely on artificially clean pools of source adapters. Stress-testing those claims on nearly 1,000 public LoRAs is timely and valuable.

- The authors do not stop at “beats prompting,” but introduce the more relevant baseline of training a target-task LoRA on exactly the same data used for coefficient tuning. That is the right comparison, and it substantially sharpens the paper’s conclusions.

- Showing that random or reinitialized LoRAs can behave similarly to selected recycled LoRAs, once a target-task LoRA is present, is a nontrivial diagnostic result that challenges the standard transfer-based narrative.  I also liked that the paper does not overclaim. The authors explicitly recover positive results in the controlled “in-house” setting, which makes the argument more credible: they are not saying adaptive merging never works, but rather that its success depends strongly on pool composition and task relevance.

- One weakness is that the paper’s central mechanistic claim remains somewhat under-supported. The regularization interpretation is plausible, but the evidence is still indirect: comparable performance from random LoRAs is suggestive, not decisive. There are other possibilities too, such as optimization effects from overparameterized coefficientized composition, or implicit ensembling-like smoothing tied to module-level scaling.

- The paper sometimes conflates two different claims: that public LoRA pools are heterogeneous/noisy, and that positive transfer is intrinsically rare. The current experiments strongly support the first claim, but only partially support the second. It could still be that the relevant signal exists in the hub and current selection methods are simply too weak to find it, especially because metadata, training provenance, and task descriptions are mostly discarded here.

In general the paper is strongest as a critique of current adaptive merging pipelines, but weaker as a definitive statement about the ceiling of LoRA reuse in practice.

---

> ### Author Rebuttal · Authors · 2026-03-31
>
> We appreciate your insightful comments and feedback – we've addressed each point below.
>
> ## 1. Regularization effect
> > "The regularization hypothesis is central to the paper’s interpretation. Can you test it more directly?"
>
> This is a great suggestion. We tested standard regularization techniques applied directly to target-task LoRA training. The table above compares the LoRA baseline (dropout 0.1, weight decay (WD) 0, batch size 8) against variants with increased dropout, weight decay on LoRA matrices, and reduced batch size, alongside adaptive merging results for reference.
>
> | Method | Pos. 5+% | Pos. 1–5% | Close ±1% | Neg. 1–5% | Neg. 5+% | Plain Avg. |
> |---|---|---|---|---|---|---|
> | LoRA Baseline | 0 | 0 | 62 | 0 | 0 | 0.657 |
> | Dropout 0.2 | 4 | 14 | 29 | 9 | 6 | 0.655 |
> | Dropout 0.3 | 3 | 13 | 32 | 10 | 4 | 0.656 |
> | Dropout 0.5 | 5 | 16 | 25 | 11 | 5 | 0.659 |
> | WD 0.01 | 1 | 10 | 31 | 12 | 8 | 0.647 |
> | WD 0.1 | 1 | 18 | 30 | 11 | 2 | 0.657 |
> | Batch Size 4 | 4 | 16 | 21 | 8 | 13 | 0.644 |
> | Batch Size 2 | 4 | 16 | 15 | 13 | 14 | 0.626 |
> | LoRA + eval. (30) | 9 | 26 | 23 | 3 | 1 | 0.675 |
> | LoRA + reinit. (30) | 8 | 25 | 24 | 4 | 1 | 0.672 |
>
> Different regularization mechanisms produce qualitatively different effects on the task-level distribution: Dropout is the most stable, preserving the overall average with marginal gains (0.655–0.659). Weight decay is more mixed, with WD 0.01 hurting overall (0.647) and WD 0.1 matching the baseline (0.657). Reducing batch size degrades performance most aggressively (0.644, 0.626), leading to many strong negative transfers.
>
> None of these standard regularizers consistently replicate the aggregate gains from adaptive merging (0.672–0.675). The merging procedure appears to provide a more effective form of regularization — potentially through the overparameterized coefficient space or implicit ensembling — that standard techniques do not capture. That said, the fact that reinitialized LoRAs perform comparably to curated ones (0.672 vs. 0.675) still suggests this benefit is closer to a structural regularization effect than genuine cross-task transfer.
>
> ## 2. Extension to different model family
>
> > "How much of the conclusion depends on the base model family?"
>
> To fully address your question, we have begun running our experiments end-to-end using the Qwen/Qwen3-4B-Instruct-2507 base model. The end-to-end replication is still running, as it involves scraping all available hub experts (near 2000 for the new base model), evaluating the hub models on the 62 downstream tasks, fine-tuning target-task LoRAs, and so on. While we believe our findings will hold across different base models, we aim to be able to share concrete results by the end of the discussion period!
>
> ## 3. Low-data setting
>
> > "…it leaves open whether the conclusions hold in very low-shot or more data-rich conditions."
>
> While rerunning the experiments in a higher data budget setting would deviate from our realistic data-constrained setting, we agree findings in lower-shot setup would be interesting. We’ve run the experimental suite with 10 samples with some modifications (e.g., no separate validation set for training; hub LoRAs evaluated on 10 samples only). We share preliminary result using “Ours” with 30 LoRAs, using random, evaluation, and reinitialized selection, w/o and w/ target LoRA in the pool:
>
> |Method|10 data|100 data (from the paper, Fig 3)|
> |-|-|-|
> |Reinit.|0.563|0.609|
> |Rand.|0.578|0.647|
> |Eval.|0.576|0.650|
> ||||
> |LoRA+reinit.|0.580|0.672|
> |LoRA+rand.|0.570|0.675|
> |LoRA+eval.|0.566|0.675|
> ||||
> |LoRA baseline|0.545|0.654|
>
> 10-data setting introduces several brittleness. Evaluation-based selection is less robust, tuning is prone to overfitting, and target-task LoRA is weaker. Hub LoRAs (rand., eval.) still outperform randomly initialized LoRAs (reinit.), but including the target LoRA (“LoRA+”) no longer consistently helps. We will share additional results (varying the pool size, evaluating other adaptive methods in this setup) once available.

---

> > ### Author Rebuttal · Reviewer_tmrT · 2026-04-03
> >
> > Thanks for the response.

---

> > > ### Author Response · Authors · 2026-04-05
> > >
> > > Our experiments using the **Qwen baseline** have finished running, and we report the two key results comparing merging methods (Table 2) and the effectiveness of the selection method (Figure 3). We will add these findings to the paper as well. We thank the reviewer again for their time and suggestion!
> > >
> > > We sampled a total of 1956 LoRA modules from the Hugging Face hub for Qwen3-4B-Instruct-2507 after filtering (details in Section 4.1). Our experiment setup remains the same, with the same compute, data budget (80 for training, 20 for validation), and hyperparameters for merging weights tuning. For any methods using randomly selected LoRAs, we sample LoRAs using three random seeds and report the averaged accuracy.
> > >
> > > ## 1. Difference across merging methods
> > >
> > > | Method | w/o target | w/ target |
> > > |---|---|---|
> > > | TIES (30) | 0.578 | 0.577 |
> > > | TSV (30) | 0.578 | 0.586 |
> > > | Simple Avg (30) | 0.569 | 0.583 |
> > > | LoraHub (30) | 0.542 | 0.636 |
> > > | Adamerge (30) | 0.583 | 0.639 |
> > > | Pi-tuning (20) | 0.253 | 0.258 |
> > > | **Ours (30)** | 0.657 | 0.681 |
> > > *Prompting baseline*: 0.578
> > > *Fine-tune baseline*: 0.668
> > >
> > > **Findings** We still find that 1) adaptive merging outperforms non-adaptive merging methods in settings both without and with the target-task LoRA, 2) the gap among methods is notably decreased once the fine-tuned LoRA is included, and 3) gradient-based tuning (Adamerge, Ours) leaves a gap over gradient-free selection (LoraHub). "Ours" in this setup outperforms other adaptive merging methods considered.
> > >
> > > **Difference** One notable difference is the decreased $\pi$-tuning performance. For $\pi$-tuning, we use the same learning rates (1e-4, 1e-5) and report the averaged performance as in the original Llama experiments, but observe a decreased overall performance. Upon investigation, we find that the “Quasi-FIM” selection method (see Section 3 and Appendix C, $\pi$-Tuning for details) used for $\pi$-tuning LoRA selection is biased toward hub LoRAs that make larger changes in the model, with the average L2 norm of Quasi-FFIM LoRAs is 2+ times higher than those of evaluation-based or random LoRAs, which leads to more severe interference. Simple averaging using Quasi-FIM LoRAs also leads to poor performance. This highlights the practical challenge of finding task-relevant LoRAs in the wild using $\pi$-Tuning when the original training data for hub LoRAs is unavailable.
> > >
> > > ## 2. Reinit. vs. Random vs. Eval. experts, w/o and w/ target-task LoRA
> > >
> > > As in Section 5.4, we examine the impact of LoRA selection across varying numbers of LoRAs in the pool ($k$), without and with the target-task LoRA in the pool. Interestingly, we find that reinitialized LoRA (“Reinit.”) performance in the Qwen3-4B-Instruct-2507 setting has stronger performance than it did for Llama 3.1 8B-Instruct setting. However, we still find that the effect of selection decreases once the target-task LoRA is included, with the performance of reinitialized LoRAs and evaluation-based (“Eval.”) LoRAs matching very closely.
> > >
> > > | Method | k=5 | k=10 | k=20 | k=30 |
> > > |---|---|---|---|---|
> > > | Reinit. | 0.628 | 0.635 | 0.644 | 0.650 |
> > > | Rand. | 0.604 | 0.615 | 0.630 | 0.634 |
> > > | Eval. | 0.647 | 0.648 | 0.657 | 0.657 |
> > > | | | | | |
> > > | LoRA+reinit. | 0.682 | 0.683 | 0.682 | 0.684 |
> > > | LoRA+rand. | 0.678 | 0.681 | 0.682 | 0.682 |
> > > | LoRA+eval. | 0.682 | 0.684 | 0.683 | 0.681 |
> > > *LoRA baseline*: 0.668

---

### Official Review · Reviewer_Emuu · 2026-03-17

**Soundness:** 3
**Presentation:** 3
**Significance:** 3
**Originality:** 3
**Overall Recommendation:** 5
**Confidence:** 3

**Summary:**

This paper studies how adaptive LoRA merging techniques perform in realistic settings, specifically when applied to thousands of LoRAs in the wild on Hugging Face Hub. The authors show that, while adaptive merging can outperform the base model, it generally fails to surpass simply training a task-specific LoRA on the same data. Surprisingly, the choice of which LoRAs to merge has little impact, and even randomly initialized LoRAs yield similar performance. This suggests that the gains mainly stem from a regularization effect rather than meaningful cross-task knowledge transfer. Positive transfer occurs only when highly relevant LoRAs are available, which is rare in realistic settings, highlighting a gap between prior research setups and real-world applicability.

**Compliance With Llm Reviewing Policy:**

Affirmed.

**Final Justification:**

I have no major concerns regarding this work, and the authors’ responses are informative and do not raise new issues. I therefore maintain my positive rating.

**Key Questions For Authors:**

Are there any insights regarding the merging of full models? The experimental design appears to be directly adaptable to full-model settings.

**Limitations:**

yes

**Strengths And Weaknesses:**

Strengths:
* The research problem of analyzing the realistic performance of LoRA merging in the wild is novel and interesting.
* The experimental design, involving 1,000 LoRAs from the Hugging Face Hub, is convincing.
* The results and conclusions are well aligned. The regularization-based explanation is reasonable.

Weaknesses: I don't see obvious weaknesses.

---

> ### Author Rebuttal · Authors · 2026-03-31
>
> We appreciate your comments and positive feedback! We addressed your question below:
>
> > “Are there any insights regarding the merging of full models? The experimental design appears to be directly adaptable to full-model settings.”
>
> This is an interesting question. The candidate expert models in our experiments are publicly available LoRAs on Hugging Face, not fully fine-tuned models. However, our setup can in theory be replicated in full model setting, as the LoRA modules merged with the base model weights is equivalent to a full model. We hypothesize that our main findings would still hold: 1) selection methods that include more task-relevant models are more beneficial, 2) certain design choices matter (e.g. gradient-based being more optimal than gradient-free alternative), and 3) if a strong model fine-tuned directly on the task data is included in the merging pool, gains from merging design choices may be marginal.
>
> In practice, however, replicating these experiments with fully fine-tuned models would be expensive, as GPU memory requirement scales rapidly with the number of models in the merging pool. Each additional model in the merging pool require at least a full model-size memory: for an 8B model in bfloat16, merging over 30 models would require at least 16G * 30 = 480GB of GPU memory.

---

> > ### Author Rebuttal · Reviewer_Emuu · 2026-04-06
> >
> > Thank you to the authors for their thoughtful response! I’ll maintain my positive rating.

---

### Official Review · Reviewer_D2PC · 2026-03-18

**Soundness:** 3
**Presentation:** 3
**Significance:** 3
**Originality:** 2
**Overall Recommendation:** 4
**Confidence:** 4

**Summary:**

This paper explores the challenge of merging LoRAs when their inventory consists of thousands of “in-the-wild” adapters obtained from online hubs. Under this setup, the Authors compared different merging strategies and found that while adaptive merging strategies may enhance the base model, their effect is comparable to training a new LoRA on a few target examples. In addition, they find that random subsets of LoRAs (as well as randomly initialised LoRAs) are as effective as selectively chosen subsets of pre-trained LoRAs once the LoRA trained on target data is included in the pool, suggesting that merging acts more as a regulariser rather than facilitating task knowledge transfer/recombination.

**Compliance With Llm Reviewing Policy:**

Affirmed.

**Final Justification:**

While some of my concerns were addressed, others remain. For instance, I still believe that the paper should have explored the mismatch between the results of hub vs in-house LoRA inventories more in depth.

I appreciate that the Authors are willing to explore the integration of Arrow as a baseline. Zero-shot routing without including the target task LoRA may be superior in settings where target task datapoints are very few, hence training a new LoRA on them becomes unreliable (as proven by the new experiments).

**Key Questions For Authors:**

- What happens if instead of a 100-shot setting, you operate in a setting with substantially fewer training examples?
- After reading the paper, I am still uncertain about the key take-away message: is it the case that adaptive merging fails in more realistic scenarios (and only succeeds on artificial, in-house pools), or rather that the quality of LoRAs available on the hub is very poor / incompatible with established evaluation tasks (hence custom, high-quality pools should be encouraged)? I’d like to see you disentangle (i) the ability of LoRA merging strategies to enable knowledge transfer and (ii) the effect of train task / eval task divergence. The latter could be controlled in a more fine-grained manner rather than just contrasting in-the-wild vs exact train-eval overlap in §5.5.

**Limitations:**

No discussion of the limitations is provided.

**Strengths And Weaknesses:**

**Strengths**
- The main novelty of this work is merging LoRAs in a realistic setting where a large pool of adapters is sourced from online hubs.
- There is a wide array of strategies for LoRA merging considered in the experiments (albeit not fully comprehensive, see below). I appreciated how the authors systematised the comparison between these strategies by breaking them down into different dimensions (selection, tuning, granularity, activation).
- The results help remind the community that simple baselines (e.g. training an individual LoRA on the few target examples used for coefficient tuning) are often neglected but can sometimes outperform significantly more complex methods.

**Weaknesses**
- The findings are substantially weakened by the fact that adapter merging strategies ignore a substantial portion of the literature, including:
    - More expressive merging, such as PHATGOOSE (Muqeeth et al. 2024), where top-k LoRAs are chosen per-token and per-layer. More expressive adaptive merging was shown to outperform strategies considered in this paper.
    - Zero-shot routing strategies, such as Arrow proposed in “Towards Modular LLMs by Building and Reusing a Library of LoRAs” (Ostapenko et al. 2024). Since no extra compute/data is necessary even though selection is adaptive, Arrow is more sample-efficient than training a new LoRA.
- Some experimental design choices are a bit questionable: for instance, why are 30 LoRAs chosen randomly for TIES, TSV, and simple averaging, when evaluation-based selection is claimed to be best based on “preliminary experiments” (line 171)? The fact that they underperform is unsurprising given this choice.
- Currently, the results focus only on accuracy; however, the selection / coefficient tuning / granularity choices all imply different levels of inference efficiency. The paper would vastly benefit from including considerations of accuracy–efficiency trade-offs when studying such large LoRA pools.

**Minor**
- Incorrect bibliography formatting (e.g., NANDA 2025)

---

> ### Author Rebuttal · Authors · 2026-03-31
>
> We appreciate your insightful comments and feedback. We've addressed each point below.
>
> ## 1. Design decisions
>
> > “… adapter merging strategies ignore a substantial portion of the literature…”
>
> ARROW and PHATGOOSE recycle LoRAs into MoE models, rather than aim for our setting, i.e. improving performance on a specific target task when data is available. In practice, they tend to underperform the best expert in the pool (commonly dubbed “Oracle” routing). As we observe the oracle LoRA typically underperforms the target-task LoRA, we anticipate ARROW and PHATGOOSE would attain relatively poor performance here. PHATGOOSE is also incompatible, as it needs the experts’ original training data.
>
> We are working on adapting ARROW to our problem setting through two variations: (a) ARROW routing on selected subsets, and (b) further tuning this router on target-task examples. We will share them by the end of the rebuttal period.
>
> > “why are 30 LoRAs chosen randomly for TIES, TSV, and simple averaging …”
>
> Thanks for bringing this up. As non adaptive merging methods are data-free, we default to random selection. We also ran experiments using evaluation-based selection across all tasks for the three methods. Below, “random” refers to the original results, “eval” to the new results with evaluation-based selection; "w/" and "w/o" to the settings with and without the target-task LoRA respectively:
>
> |Method|random (w/o)|random (w/)|eval (w/o)|eval (w/)|
> |-|-|-|-|-|
> |Simple Avg.|0.496|0.536|0.543|0.577|
> |TIES|0.467|0.466|0.466|0.463|
> |TSV|0.488|0.522|0.504|0.502|
>
> While performance improves for the simple averaging method, overall these methods continue to underperform adaptive merging methods that tune merging coefficients using task data points (see Table 2). We will add this result into our paper.
>
> ## 2. Efficacy analysis
>
> > “…considerations of accuracy–efficiency trade-offs when studying such large LoRA pools.”
>
> This is a great point -- we summarized the cost of each method, broken down by # trainable params, selection, and tuning approach.
>
> O(E): LoRA size;
> k: # selected LoRAs;
> K: total # available LoRAs;
> L: # layers in the base model;
> M: # target modules
>
> |Method|Trainable params|LoRA selection|Tuning steps|
> |-|-|-|-|
> |Simple averaging|0|O(1)|0|
> |TSV|0|O(1)|0|
> |TIES|0|O(1)|0|
> |LoraHub|k|O(1)|100|
> |AdaMerging|k × L × M|O(1)|100|
> |π-tuning|k × L × M + k × O(E)|K evals|100|
> |Ours|k × L × M|K evals|100|
> |Target-task LoRA|O(E)|N/A|100|
>
> Adaptive merging methods are more expensive but consistently outperform non-adaptive methods. However, once the target-task LoRA is in the pool, the efficiency-accuracy trade-off observed in the previous setting no longer holds (Fig. 3 & Appendix D, Fig. 9). We will include this cost breakdown in our paper.
>
> ## 3. Low-data setting
>
> > “... with substantially fewer training examples?”
>
> As you suggested, we’ve begun replicating our experiments with 10 samples (no train/val split, selection based on 10 samples). We share early result using “Ours” with 30 LoRAs, across random, evaluation, and reinitialized, w/o and w/ target LoRA:
>
> |Method|10 data|100 data|
> |-|-|-|
> |Reinit.|0.563|0.609|
> |Rand.|0.578|0.647|
> |Eval.|0.576|0.650|
> ||||
> |LoRA+reinit.|0.580|0.672|
> |LoRA+rand.|0.570|0.675|
> |LoRA+eval.|0.566|0.675|
> ||||
> |LoRA baseline|0.545|0.654|
>
> Hub LoRAs (rand., eval.) still outperform randomly initialized LoRAs (reinit.), but including the target LoRA (“LoRA+”) no longer consistently helps, possibly due to brittleness of 10-data setting (e.g., selection being less robust, risk of overfitting, weaker target-task LoRA). We will share additional results (varying the pool size, evaluating other adaptive methods in this setup) once available.
>
> ## 4. Clarifications
> > “is it the case that adaptive merging fails in more realistic scenarios …, or rather that the quality of LoRAs available on the hub is very poor …?”
>
> You’re right that there are two (potentially overlapping) arguments:
>
> Adaptive merging methods achieve positive transfer using hub LoRAs (Figure 2.a and 2.b), but extracting additional gains with the target-task LoRA included in the pool is difficult. This may be due to the methods’ limitation in extracting transferable knowledge from hub LoRAs given the strong signal from the target-task LoRA.
>
> However, adaptive merging shows positive transfer when hub LoRAs is replaced with carefully selected “in-house” LoRAs. This suggests either 1) a different selection method is necessary, as several hub LoRAs exhibit potential for positive transfer (Figure 1), or 2) the hub models are incompatible with target-task LoRAs due to factors such as training methods or data, though isolating these causes are challenging given limited hub LoRA metadata.
>
> We will clarify these possibilities explicitly in our conclusion, as we see both directions as meaningful paths for followup work. Please let us know if you have additional suggestions!
>
> > “Incorrect bibliography formatting”
>
> Thanks for catching this!

---

> > ### Author Rebuttal · Reviewer_D2PC · 2026-04-03
> >
> > Thank you for your rebuttal. While some of my concerns were addressed, others remain. For instance, I still believe that the paper should have explored the mismatch between the results of hub vs in-house LoRA inventories more in depth.
> >
> > I appreciate that the Authors are willing to explore the integration of Arrow as a baseline. Zero-shot routing without including the target task LoRA may be superior in settings where target task datapoints are very few, hence training a new LoRA on them becomes unreliable (as proven by the new experiments).
> >
> > I think the efficiency table and the 10-shot adaptation results provided in the rebuttal are quite insightful and should be included in the next version of the paper.

---

> > > ### Author Response · Authors · 2026-04-06
> > >
> > > We thank the reviewer for their suggestion and continued engagement. We have some interesting results from ARROW and data-scarce regime results that we share below.
> > >
> > > ## 1. ARROW results
> > >
> > > Our first set of experiments follow the same setting as merging methods comparison (Table 2). Here, we test out four variants of ARROW method, differ by selection method and whether we tune the router on target data. Following the original paper, we activate 4 experts at each token. When training the router, we use lr=1e-2 for up to 400 steps.
> > >
> > > As a non-adaptive method, ARROW has its performance consistent with other non-adaptive methods. With random selection (0.471), it is comparable to the prompting baseline (0.467), and with eval-based selection (0.564), it falls in range with TIES, TSV, and Simple Avg. Eval-based selection helps consistently (0.564 vs 0.471 w/o target; 0.639 vs 0.613 w/ target), aligning with the trend among non-adaptive methods.
> > >
> > > Once the router is tuned on target-task data, making it adaptive, performance improves dramatically — Arrow tuned (eval) reaches 0.648 w/o target and 0.670 w/ target, approaching Adamerging (0.650/0.675) and Ours (0.652/0.675). The gap between eval and random selection nearly vanishes after tuning (0.648 vs 0.647 w/o target; 0.670 vs 0.668 w/ target), echoing our finding in Section 5.4 that target-task data reduces the importance of selection. However, tuned Arrow does not surpass Ours or Adamerging in the w/ target setting (0.670 vs 0.675).
> > >
> > > | Method | w/o target | w/ target |
> > > |-|-|-|
> > > | TSV (30) | 0.488 | 0.522 |
> > > | TIES (30) | 0.467 | 0.466 |
> > > | Simple Avg (30) | 0.496 | 0.536 |
> > > | Simple Avg (all) | 0.495 | 0.496 |
> > > | Adamerging (30) | 0.650 | 0.675 |
> > > | π-Tuning (20) | 0.668 | 0.668 |
> > > | LoraHub (30) | 0.563 | 0.663 |
> > > | Ours (30) | 0.652 | 0.675 |
> > > | | | |
> > > | ARROW (eval 30) | 0.564 | 0.639 |
> > > | ARROW (random 30) | 0.471 | 0.613 |
> > > | ARROW tuned (eval 30) | 0.648 | 0.670 |
> > > | ARROW tuned (random 30) | 0.647 | 0.668 |
> > > | | | |
> > > | Prompting | 0.467 | - |
> > > | Fine-tune (LoRA) | - | 0.657 |
> > >
> > >
> > > ## 2. Data scarce regime
> > >
> > > We completed the data scarce regime experiments and included ARROW variants in the comparison.
> > >
> > > **Merging methods comparison**
> > >
> > > The overall performance of adaptive methods is notably lower in 10 data settings than in the 100 data setting.
> > > Comparing across adaptive merging methods in the 10 data setting, we find that including target-task LoRA isn’t always consistently helpful as it has in the 100 data setting. ARROW variants show remarkable results in this setting, especially when the router is further tuned on downstream task data, this shows the importance of clever initialization at the data scarce regime. And worth notingly, when the router is tuned, the gap between eval based selection and random selection is small, suggesting the improvement also doesn’t come from finding highly relevant hub models.
> > >
> > > | Method | w/o target | w/ target |
> > > |-|-|-|
> > > | LoraHub (30) | 0.517 | 0.587 |
> > > | Adamerging (30) | 0.558 | 0.557 |
> > > | $\pi$-Tuning (20) | 0.546 | 0.549 |
> > > | Ours (30) | 0.578 | 0.566 |
> > > | Prompting | 0.467 | - |
> > > | Fine-tune (LoRA)| - |  0.546 |
> > > | | | |
> > > | ARROW (eval 30) | 0.550 | 0.575 |
> > > | ARROW (random 30) | 0.471 | 0.574 |
> > > | ARROW tuned (eval 30) | 0.595 | 0.583 |
> > > | ARROW tuned (random 30) | 0.591 | 0.588 |
> > > | | | |
> > >
> > > **Selection method**
> > >
> > > The gap among selection methods (reinitialized vs. randomly selected vs. evaluation-based LoRAs) narrows after including the target-task LoRA. Interestingly, unlike in the 100 data sample setting, increasing the number of LoRAs in the pool (k) does not always lead to higher performance when the target-task LoRA is included.
> > >
> > > |Method| k=5 | k=10 | k=20 | k=30 |
> > > |-|-|-|-|-|
> > > |Reinit.| 0.537 |0.543 |0.560 |0.567 |
> > > |Rand.| 0.558 | 0.571 | 0.579 | 0.575 |
> > > |Eval.|0.574 | 0.574 | 0.577 | 0.578 |
> > > ||||
> > > |LoRA+reinit.|0.585| 0.585 |0.584 | 0.582 |
> > > |LoRA+rand.|0.582 |0.579 |0.574 |0.569 |
> > > |LoRA+eval.|0.584 |0.578 |0.570 |0.566 |
> > > ||||
> > > |LoRA baseline|0.545|
> > >
> > > We will include these results in the paper as well. If we have sufficiently addressed your concern, we would appreciate if you considered raising your score!

---

### Decision · Program_Chairs · 2026-04-30

**Decision:**

Accept (regular)

**Comment:**

Summary: This paper tests adaptive LoRA merging methods using a pool of 1000 LoRAs for Llama 3.1 8B-Instruct sourced from the Hugging Face Hub. The authors systematically compare adaptive and non-adaptive merging strategies. The paper reports three key findings: (1) adaptive merging improves over the base model, but typically does not surpass training a target-task LoRA on the same small dataset, (2) once the target task LoRA is included, the choice of which hub LoRAs to merge has negligible impact, and (3) positive transfer reliably occurs only when highly relevant LoRAs are available.

Strengths: Reviewers all praised the systematic and comprehensive analysis using the full 1000 LoRAs. Reviewers also found the comparison with a target task LoRA surprising and important for the community. Collectively the paper presents practical guidance for practitioners on merging methods while also presenting strong evidence of a potentially negative result on merging.

Weaknesses: Some reviewers felt that the regularization interpretation, while plausible, is not directly verified. Experiments show standard regularizers do not replicate gains from merging, but did not provide full conclusion on what other effect is provided. The provided analysis also does not fully characterize the LoRAs, and it remains inconclusive how noisy and heterogenous the actual pool of LoRAs is.

**Final Recommendation: Accept**

Justification: The paper provides a well-executed empirical study with important and practically relevant findings for the LoRA merging community. While there are open questions regarding the regularization effect, and pool of LoRAs, the results are sufficiently interesting already, and the systematic comprehensive analysis is of great importance to the community and practitioners more broadly. During the rebuttal, the authors provided more conclusive evidence as well for a Qwen model corroborating the results with an additional model.